# Curcumin Derivative CU4c Exhibits HDAC-Inhibitory and Anticancer Activities Against Human Lung Cancer Cells In Vitro and in Mouse Xenograft Models

**DOI:** 10.3390/ph18070960

**Published:** 2025-06-26

**Authors:** Narissara Namwan, Gulsiri Senawong, Chanokbhorn Phaosiri, Pakit Kumboonma, La-or Somsakeesit, Pitchakorn Sangchang, Thanaset Senawong

**Affiliations:** 1Department of Biochemistry, Faculty of Science, Khon Kaen University, Khon Kaen 40002, Thailand; narissaranamwan@kkumail.com (N.N.); gulsiri@kku.ac.th (G.S.); pitchakorn_sa@kkumail.com (P.S.); 2Department of Chemistry, Faculty of Science, Khon Kaen University, Khon Kaen 40002, Thailand; chapha@kku.ac.th; 3Department of Applied Chemistry, Faculty of Science and Liberal Arts, Rajamangala University of Technology Isan, Nakhon Ratchasima 30000, Thailand; pakit.ku@rmuti.ac.th; 4Department of Chemistry, Faculty of Engineering, Rajamangala University of Technology Isan, Khon Kaen 40000, Thailand; laor.so@rmuti.ac.th

**Keywords:** lung cancer, cisplatin, gemcitabine, curcumin derivative CU4c, drug combination, cell cycle arrest, apoptosis, HDAC inhibitor, chemosensitizer

## Abstract

**Background/Objectives**: Drug resistance and severe side effects caused by gemcitabine (Gem) and cisplatin (CDDP) are common. This study aimed to investigate the combined effects of CU4c and Gem or CDDP on lung cancer cells in vitro and in nude mouse xenograft models. **Methods**: Antiproliferative activity and drug interaction were evaluated using MTT and Chou–Talalay methods, respectively. Apoptosis induction and cell cycle arrest were analyzed by flow cytometry. The expression levels of proteins were evaluated by Western blot analysis. The HDAC-inhibitory activity of CU4c was confirmed in vitro, in silico, and in A549 cells. **Results**: CU4c inhibited the proliferation of A549 cells in a dose- and time-dependent manner but had little effect on the growth of noncancerous Vero cells. CU4c synergistically enhanced the antiproliferative activities of CDDP (at 24 h) and Gem (at 48 and 72 h) against A549 cells. Combined CU4c and CDDP notably inhibited A549 proliferation by triggering cell cycle arrest at S and G2/M phases at 24 h with elevated levels of p21 and p53 proteins. Combined CU4c and Gem induced cell cycle arrest at both the S and G2/M phases at 48 h via upregulating the expression of the p21 protein. CU4c enhanced the apoptotic effects of CDDP and Gem by increasing the Bax/Bcl-2 ratio, pERK1/2, and Ac-H3 levels. Combined CU4c and Gem significantly reduced tumor growth while minimizing visceral organ damage in animal study. **Conclusions**: These results suggest that CU4c enhances the anticancer activity of CDDP and Gem and reduces the toxicity of Gem in animal studies.

## 1. Introduction

Lung malignancies are the foremost cause of cancer-related deaths globally, accounting for 11.4% of total cases. They have a poor prognosis, with rates of survival over five years ranging from 4% to 17%, depending on the stage at diagnosis [1]. Most lung cancer cases involve non-small cell lung cancer (NSCLC), which accounts for more than 80% of the total lung cancer cases reported. Most patients are diagnosed with NSCLC at the metastatic stage, making surgical intervention unsuitable [2]. Consequently, chemotherapy is the most suitable therapeutic option for treating NSCLC [3]. Chemotherapy medications frequently employed to treat patients with lung cancer include gemcitabine (Gem), cisplatin (CDDP), and vinorelbine (VIN) [4]. Nonetheless, medication resistance and significant negative outcomes with clinical trials are prevalent and restrict their application [5]. To prevent drug resistance, various pharmaceuticals are often recommended together as a combination treatment. Synergistically combining medications effectively eliminates cancer cells at lower doses, decreasing drug resistance and adverse effects for patients [4]. Accordingly, the search for novel medication combinations or chemotherapy sensitizers to enhance effectiveness and minimize side effects has become a key priority [6].

Epigenetic modifications are key factors in promoting tumor growth. Cancer cells exhibit an altered acetylation profile, which is essential for tumor progression and contributes to a reduced response to lung cancer treatments [7]. Histone deacetylases (HDACs) are enzymes that play a crucial role in chromatin remodeling. They remove acetyl groups from acetylated histones, which are essential for structuring and compacting DNA within cells [8]. HDACs possess several biological activities in both health and illness by modulating diverse cellular processes, including cell proliferation, cell cycle, survival, and apoptosis [9]. Moreover, the reactivation of tumor-suppressor genes that have been epigenetically repressed depends on HDACs [10]. An accumulation of evidence suggests that abnormal expression or activation of HDACs facilitates carcinogenesis as well as assisting in the emergence of resistance to lung cancer medications [11]. Recent clinical studies have demonstrated that HDAC inhibitors (HDACis) can improve the antitumor efficacy of molecularly targeted medications and cisplatin-based chemotherapy in patients with lung cancer [12,13]. To mitigate adverse effects and attain a more comprehensive approach to treatment, current research is dedicated to improving HDACis derived from natural sources, as well as the exploration of combination therapies and additive medications [14]. Thus, the development of novel HDACis for cancer treatment has received considerable research attention [15,16].

Among anticancer drugs, natural products and their structural analogs are attractive candidates for pharmacotherapy. Curcumin (CU) is a compound present in the rhizome of the Curcuma genus plants, most notably in turmeric (*Curcuma longa* L.). It has been reported to possess numerous biological activities, including antimicrobial, antioxidant, anticancer, and antiangiogenic properties [17,18]. Additionally, curcumin has been characterized as an HDACi, which reduces the expression of HDAC classes 1, 2, 3, 4, 5, 6, 8, and 11 in mice and various cancer cell lines [19]. Prior results showed that CU caused cytotoxicity in NSCLCs by inhibiting cell proliferation and differentiation, promoting apoptosis, halting the cell cycle at the G2/M phases, and generating reactive oxygen species (ROS). This series of actions ultimately activate the DNA damage signaling system [20,21]. Nonetheless, the medical utilization of CU is limited by its inadequate water solubility, fast metabolism, and elevated toxicity in noncancerous cells. Consequently, considerable attention has been paid to the exploration of novel chemically synthesized curcumin analogs exhibiting comparable biological activity [22]. Currently, numerous studies have emerged on the antiproliferation effects of curcumin analogs or derivatives in various cancer cell lines [23]. For example, the curcumin derivatives CA-5f and 5k (3,4-dichlorobenzyltriazole methyl curcumin) demonstrated stronger suppression of lung cancer cells’ growth compared to curcumin, without causing significant harm to normal cells [24,25]. Furthermore, our previous research indicated that the curcumin derivative CU17 exhibited cytotoxic effects against NSCLC cells, demonstrating anticancer efficacy through G2/M phase arrest and apoptotic mechanisms [20].

Recently, fifty-nine curcuminoid derivatives were synthesized with different side chains at the phenolic moiety, with lower physicochemical instability but stronger HDAC-inhibitory activities than CU [26]. Therefore, we hypothesized that a curcumin derivative CU4c (CU4c) with ethyl groups at the side chains (Figure 1) would exhibit HDAC-inhibitory and anti-tumor activities against non-small cell lung cancer (NSCLC) cells because CU4c presented more potent HDAC inhibition (87%) than CU (67%) [26]. This study aimed to further investigate HDAC-inhibitory activity (in vitro and in silico) in more detail and the anticancer activity of CU4c in single and combination treatments, as well as explore its underlying anticancer mechanism against human lung cancer A549 cells both in vitro and in nude mouse xenograft models.

## 2. Results

### 2.1. HDAC-Inhibitory Activity of the Curcumin Derivative CU4c In Vitro and in A549 Cells

The inhibitory activity of CU4c was determined using the Fluor-de-Lys HDAC Fluorometric Activity Assay Kit. CU4c inhibited the activity of the HDAC enzyme in a dose-dependent manner, showing the half-maximal inhibitory concentration (IC_50_) value of 18.75 ± 0.23 µM (Figure 2a). In addition, a Western blot analysis was conducted to investigate the HDAC-inhibitory effect of CU4c in A549 cells by measuring the acetylation of histone H3. CU4c treatment triggered hyperacetylation of histone H3 in A549 cells in a dose-dependent manner, and a significant accumulation of acetylated histone H3 could be observed with CU4c at 50 and 100 µM (Figure 2b,c).

### 2.2. HDAC-Inhibitory Activity of the Curcumin Derivative CU4c In Silico

A molecular docking study was also conducted to support the in vitro results. Molecular docking was performed, employing CU4c as a ligand, docking with the crystal structure of human class I HDACs (HDAC1, 2, 3, and 8) and class II HDACs (HDAC4, 6, and 7) (Figure 3). The CU4c showed the lowest binding energy (ΔG) and inhibitory constant (Ki) values for all Class I and II HDACs (Table 1). The binding energies ranged from −5.89 to −7.92 kcal/mol. The highest binding energy was established with HDAC VII, resulting in a binding energy of −7.92 kcal/mol (Table 1). Molecular modeling of CU4c with HDAC class I and II showed that it would probably interact with several amino acids in the enzymes’ active regions. The CU4c-HDAC1 complex formed strong hydrogen bonds with Ser148 (2.50 Å) and Phe205 (2.46 Å), as well as weaker connections with Gly146 (2.10 Å), Ser148 (2.50 Å, 3.05 Å), Asp181 (2.92 Å, 2.94 Å), and Pro206 (2.55 Å). In addition, CU4c interacts with Tyr204 (4.09 Å, 5.24 Å) and Pro206 (4.56 Å) via π-interaction at the active site of HDAC1 (Figure 3a). The HDAC2 binding site included a zinc ion chelation (2.82 Å), one weak hydrogen bond, and seven π-interactions with CU4c, including Gly154 (2.70 Å), Pro106 (4.10 Å, 4.66 Å), His146 (3.98 Å), Phe155 (4.31 Å), His183 (4.69 Å), and Phe210 (4.35 Å, 4.85 Å), respectively (Figure 3b). The binding of CU4c and HDAC3 at the active site involved zinc ion chelation (2.96 Å), two strong and three weak hydrogen bonds, and seven π-interactions with residues such as Tyr198 (2.18 Å), Cys268 (2.97 Å), Asp93 (2.80 Å), Asp225 (2.04 Å, 2.72 Å), Phe144 (4.24 Å), His172 (3.72 Å, 4.65 Å), Leu266 (4.62 Å), Try298 (40.40 Å), His172 (4.65 Å), and Phe200 (4.67 Å), respectively (Figure 3c). CU4c binds to HDAC4 cavity via one strong and three weak hydrogen bonds, and six π-interactions with residues Lys20 (2.12 Å), His159 (2.71 Å), Gly330 (1.92 Å, 2.76 Å), Arg37 (5.32 Å), Pro156 (5.33 Å), Phe168 (5.36 Å), His198 (4.93 Å), Phe227 (4.42 Å), and Leu229 (4.83 Å), respectively. Moreover, CU4c interacted with a 2.4 Å zinc ion in HDAC4’s catalytic site (Figure 3d). CU4c and HDAC6 have two strong hydrogen bonds, three weak hydrogen bonds, and eight π-interactions with residues Ser150 (1.84 Å), Tyr363 (2.50 Å), Asp149 (2.45 Å), Pro262 (2.31 Å), His232 (3.05 Å), His192 (4.99 Å), His193 (4.34 Å), Phe202 (4.06 Å), Trp261 (2.31 Å, 5.41 Å, 5.45 Å), and His232 (3.87 Å, 4.38 Å), respectively (Figure 3e). CU4c forms strong hydrogen bonds with the Arg731 residue (3.04 Å, 2.98 Å) in HDAC7’s active site. One weak hydrogen bond occurred between CU4c and Asp707 (2.50 Å) residue in the HDAC7 active site. The CU4c-HDAC7 complex has six π-interactions with residues Pro667 (4.73 Å), His669 (4.25 Å), His670 (4.92 Å), Phe679 (4.37 Å), His709 (4.37 Å), and Phe737 (4.58 Å). Coordination between CU4c and the zinc ion (2.20 Å) was detected at HDAC7 active sites, where the zinc ion is a cofactor (Figure 3f). CU4c binds to HDAC8 cavity via one strong hydrogen bond, two weak hydrogen bonds, and three π-interactions with residues Ala339 (1.80 Å), Gly271 (2.31 Å), Ile348 (2.71 Å), Pro273 (4.31 Å, 4.55 Å), Gly341 (4.55 Å), and Ile348 (5.08 Å), respectively (Figure 3g). The findings indicate that CU4c functions as an HDAC inhibitor, as shown by in silico docking and the in vitro study, supporting its efficacy in A549 cells.

### 2.3. Antiproliferative Effects of CU4c on Lung Cancer and Noncancer Cells

The cytotoxic effectiveness of CU4c on human lung cancer and noncancerous Vero cells was evaluated by utilizing the MTT assay. The results found that the proliferation of A549 cells was preferentially inhibited by CU4c treatment in a dose- and time-dependent manner (Figure 4). CU4c had an IC_50_ value of 93.90 ± 1.84, 60.43 ± 2.18, and 40.34 ± 1.20 µM against A549 cells after 24, 48, and 72 h of exposure, respectively (Figure 4a). In contrast, CU4c treatment did not inhibit the proliferation of noncancerous Vero cells, which exhibited IC_50_ values higher than 200 µM at exposure times of 24, 48, and 72 h (Figure 4b). The findings indicate that CU4c suppressed the growth of lung cancer cells while demonstrating nontoxicity toward noncancerous cells.

### 2.4. Selectivity Index (SI) of CU4c Against Human Lung Cancer A549 Cells

The selectivity index can be used to express a selective effect of a drug, which is determined by comparing the cytotoxic activity of the drug against a lung cancer A549 cell line with its activity against a noncancerous Vero cell line. SI values of the tested compounds are shown in Table 2. CU4c exhibited SI values of 2.13, 3.31, and 4.96 during incubation periods of 24, 48, and 72 h, respectively. The results suggest that CU4c is considered a selective agent against the NSCLC A549 cell line.

### 2.5. Antiproliferative Effects of CU4c in Combination with Standard Chemotherapeutic Drugs (CDDP and Gem) on A549 Cells

To assess whether the antitumor effects of CU4c could be enhanced by combining it with existing anticancer drugs (CDDP and Gem) against A549 cells, we conducted combination treatments using subtoxic doses of CDDP and Gem at IC_20_ and IC_30_. The IC_50_ values of CU4c in conjunction with a subtoxic dose of CDDP and Gem are presented in Table 3. The type of drug interaction was determined by using the calculated CI and DRI values based on the median-effect principle of the Chou–Talalay method [28]. The CI values for the combination treatments of CU4c and CDDP at IC_20_ against A549 cells after 24, 48, and 72 h were 0.38 ± 0.00, 1.03 ± 0.01, and 1.23 ± 0.10, respectively. These values indicated synergistic, nearly additive, and slight antagonistic effects at these time points (Table 3). The synergistic effect observed at 24 h exposure led to a 3.13-fold dose reduction for CDDP and a 16.59-fold dose reduction for CU4c. For combination treatments of CU4c and CDDP at IC_30_, a synergistic effect (CI = 0.53 ± 0.00) was noted at the 24 h exposure; however, slight antagonistic effects were observed at the 48 and 72 h exposures (CI = 1.35 ± 0.04 and 1.90 ± 0.11, respectively) against A549 cells (Table 3). The synergistic effect at 24 h resulted in a 2.01-fold dose reduction for CDDP and a 33.78-fold dose reduction for CU4c. Moreover, combination treatments of CU4c and Gem at IC_20_ against A549 cells demonstrated CI values of 0.26 ± 0.02 and 0.31 ± 0.02 at 48 and 72 h, respectively, indicating strong synergistic effects (Table 3). The dose reductions via synergistic effects at 48 and 72 h were 3.83- to 16.00-fold for Gem and 2.40- to 3.19-fold for CU4c. The CI values for the combination treatments of CU4c and Gem at IC_30_ against A549 cells at 48 and 72 h were 0.76 ± 0.01 and 0.91 ± 0.12, respectively, reflecting moderate synergistic and nearly additive effects, respectively (Table 3). The synergistic effect at 48 h yielded an 8.37-fold dose reduction for Gem and a 3.59-fold reduction for CU4c. Overall, these findings indicate that CU4c potentiated the anticancer efficacy of Gem and CDDP against A549 cells.

To demonstrate the drug interactions estimated using the Chou–Talalay approach, as illustrated in Table 3, we conducted combination treatments with CU4c at IC_50_ levels obtained from the combination treatments with IC_20_ and IC_30_ of CDDP or Gem. The efficacy of drug combination therapies between CU4c and CDDP or Gem at subtoxic doses against A549 cells was evaluated in comparison to single-drug treatments (Figure 5 and Figure 6). The combination of CU4c and CDDP at a subtoxic IC_20_ dosage significantly reduced the growth of A549 cells compared to CDDP monotherapy at 24, 48, and 72 h. Notably, the drug interactions exhibited synergistic effects only after 24 h of exposure (Figure 5a), whereas additive and antagonistic effects were observed after 48 and 72 h, respectively (Figure 5b,c). Furthermore, CU4c in conjunction with CDDP at an IC_30_ subtoxic dosage for 24, 48, and 72 h significantly more effectively inhibited the growth of A549 cells than the administration of CDDP alone. However, the drug interaction exhibited synergistic effects only during 24 h of incubation (Figure 5d), and had antagonistic effects at 48 and 72 h of exposure (Figure 5e,f). Importantly, the combined treatment of CU4c and Gem at subtoxic doses of IC_20_ and IC_30_ dramatically suppressed the proliferation of A549 cells after 48 and 72 h of exposure, in comparison with Gem monotherapy, consistent with their synergistic and slightly additive effects (Figure 6a–d).

### 2.6. Effect of Combined Therapies of CU4c with CDDP or Gem on Cell Cycle Arrest Against A549 Cells

Based on the synergistic effects of CU4c combined with CDDP or Gem against A549 cells, we further explored whether cell cycle arrest induction was associated with the efficacy of CU4c in inducing cytotoxicity in combination therapies. The combination treatment of 5.66 µM CU4c and 21.00 µM CDDP led to a significant increase in the percentage of cells arrested in the S and G2/M phases (39.60 ± 2.60% and 18.90 ± 1.90%, respectively) when compared to CDDP (30.95 ± 1.15% and 9.05 ± 0.05%, respectively) administered as single-drug treatments (Figure 7a,b). Furthermore, the combination of 2.78 µM CU4c and 32.73 µM CDDP resulted in a noticeable increase in the SubG1 phase (2.35 ± 0.35%) compared to the CDDP single-drug treatment (1.70 ± 0.00%) (Figure 7a,b). Additionally, we assessed the expression of proteins related to the induction of cell cycle arrest, specifically p53 and p21. The cotreatment with CU4c and CDDP significantly raised the levels of p53 and p21 proteins compared to both the individual treatments and the control group (Figure 7c,d). The findings indicated that combining CU4c and CDDP caused cell cycle arrest at the S and G2/M phases in lung cancer A549 cells by enhancing the expression of p53 and p21 proteins.

Additionally, the combination of 18.92 µM CU4c with 0.68 µM Gem and 16.82 µM CU4c with 1.30 µM Gem resulted in a significantly higher percentage of cells arresting in the S phase (20.60 ± 0.00% and 24.00 ± 3.00%, respectively) and G2/M phases (26.75 ± 0.25% and 29.55 ± 0.25%, respectively) compared to Gem alone, which showed S phase arrest at 15.95 ± 0.05% and 20.60 ± 3.10% and G2/M phase arrest at 13.65 ± 0.45% and 16.85 ± 3.25%, respectively (Figure 8a,b). Western blot analysis indicated that p21 protein expression was not significantly raised in the cotreatment with 18.92 µM CU4c and 0.68 µM Gem relative to single and control treatments. However, the combination of 16.82 µM CU4c with 1.30 µM Gem caused notably increased p21 expression relative to single and control treatments (Figure 8c,d). The results revealed that the combination of CU4c and Gem induced cell cycle arrest at the S and G2/M phases in A549 lung cancer cells through upregulating p21 protein expression.

### 2.7. Effect of Combined Therapies of CU4c with CDDP or Gem on the Activation of Cellular Apoptosis Against A549 Cells

CU4c had a synergistic effect on A549 cells during both CDDP and Gem combination therapies. Therefore, we investigated whether the activation of apoptosis was linked to CU4c’s ability to induce cytotoxicity in these combination therapies. Treatment with CU4c at 5.66 µM and CDDP at 21.00 µM did not significantly induce apoptosis in A549 cells compared to either agent administered alone. The combination treatment of CU4c at 2.78 µM and CDDP at 32.73 µM significantly augmented apoptotic induction in A549 cells compared to the individual treatments with CDDP or CU4c (Figure 9a,b). To investigate the apoptotic mechanism of combination treatments on A549 cells, apoptotic-related proteins were analyzed using Western blot. The expression level of the Bax protein did not alter in all treatments, while the expression level of the Bcl-2 protein decreased substantially in both CDDP monotherapies and combination therapies relative to the control treatment (Figure 9c,d). Notably, an increase in the Bax/Bcl-2 expression ratio was observed only in the treatment combining CU4c at 2.78 µM with CDDP at 32.73 µM (Figure 9e). Additionally, the combination treatment of CDDP and CU4c significantly upregulated the expression of pERK1/2 and Ac-H3 proteins in A549 cells compared to the single-drug treatments and the vehicle control treatments (Figure 9c,d).

The treatment with Gem (0.68 and 1.30 µM) or CU4c (16.82 and 18.92 µM) alone could trigger apoptosis in A549 cells; however, the combination of CU4c at 18.92 µM with Gem at 0.68 µM and 16.82 µM CU4c with 1.30 µM Gem exhibited a more significant induction of apoptosis in A549 cells than other monotherapy treatments (Figure 10a,b). The expression of the proapoptotic Bax protein remained unchanged after all treatments; however, all treatments resulted in the downregulation of antiapoptotic Bcl-2 protein expression (Figure 10c,d). Importantly, the cotreatment significantly increased the Bax/Bcl-2 expression ratio compared to other single-agent therapies (Figure 10e). Furthermore, the CU4c in conjunction with Gem markedly enhanced the expression of pERK1/2 and Ac-H3 proteins relative to individual and vehicle control treatments (Figure 10c,d). Hence, the findings established that the efficacy of CU4c in enhancing the anticancer effects of CDDP and Gem on A549 cells may be related to synergistic apoptosis induction via the upregulation of pERK1/2, Ac-H3, and the Bax/Bcl-2 ratio. Nonetheless, the combination of CU4c and Gem demonstrated a more significant inhibition of lung carcinoma A549 cell proliferation compared to the combination treatment of CU4c and CDDP. Therefore, CU4c in conjunction with Gem treatment was chosen for additional in vivo experimentation.

### 2.8. CU, CU4c, and Gem Single Treatments and Combinations of CU4c with Gem Inhibit the Growth of A549 Xenograft Tumors

We aimed to evaluate the antitumor effects of CU, CU4c, and Gem administered individually, as well as the combination of CU4c with Gem. During the acclimatization period, seven mice perished (due to an unexpected accident during transportation), resulting in a total of twenty-three mice remaining. Xenograft tumor models were produced by implanting A549 cells into BALB/cAJcl-Nu/Nu nude mice. When tumor volume reached approximately 100 mm^3^, 23 mice were randomized into five treatment groups: the vehicle control (n = 5), CU (n = 4), CU4c (n = 5), Gem (n = 4), and Gem + CU4c-15 (n = 5). The xenograft mice received intraperitoneal injections every three days for 21 days (Figure 11a). The treatments included 20% PEG400 in olive oil as a vehicle control, 30 mg/kg CU, 30 mg/kg CU4c, 50 mg/kg Gem, and a combination of 30 mg/kg CU4c with 50 mg/kg Gem. No mice died in any group after A549 cell implantation and treatment with either single or combination medications over the treatment period (Figure 11b). After completion of the therapy, the tumors were removed from the mice and then photographed (Figure 11f). Mice treated with CU and CU4c alone exhibited a reduction in tumor volume, growth, and weight compared to those treated with a vehicle control (Figure 11c–e). Importantly, mice administered the combination of CU4c and Gem demonstrated a substantial reduction in tumor growth relative to those receiving either therapy alone (Figure 11c). Additionally, the combined treatment of CU4c and Gem led to a decrease in tumor weight relative to Gem monotherapy (Figure 11d). The %TGI values were greater in the group receiving the combined treatment than in the monotherapy groups (Figure 11e). Overall, CU4c in conjunction with Gem demonstrates a synergistic effect in inhibiting tumor development in mouse models.

### 2.9. Toxicological Evaluation of CU, CU4c, and Gem Monotherapy and Combinations of CU4c with Gem in Nude Mouse Xenograft Models

Toxicological investigations were conducted in vivo using mouse xenografts, examining the effects of 30 mg/kg CU, 30 mg/kg CU4c, and 50 mg/kg Gem, both as individual treatments and in combination with CU4c. The assessments focused on %BWC, organ weight, and histopathological examination of visceral organs, including the liver, kidneys, and spleen. The %BWC in the groups treated with CU, CU4c, and the combination of Gem and CU4c showed no statistically significant differences when compared to the vehicle control group. However, a substantial reduction in body weight was observed in the Gem-treated groups compared to the vehicle control group. All treatments did not cause any significant changes in the weights of the liver and kidneys relative to the vehicle control group. Furthermore, the spleen weights in mice treated with either CU or CU4c alone were unchanged. In contrast, Gem monotherapy and its combinations resulted in significant increases in spleen weights compared to the vehicle control group (Table 4).

A histological analysis of tumor tissues from the treated groups showed a significant reduction in proliferating cells, resulting in the formation of a uniform pink region (star). Additionally, the morphological features of tumor cells in the treated groups revealed apoptotic cells (red arrow) in contrast to those in the control group (Figure 12a). Moreover, the combination treatment led to a higher incidence of cell mortality compared to the single-drug treatment groups (Figure 12a). Importantly, no significant histological changes were observed in the kidney, liver, and spleen tissues of the CU4c-treated groups compared to the vehicle control group (Figure 12b–d). In contrast, individual injections of CU and Gem showed evidence of necrosis (black arrows) in the renal tissues (Figure 12b). The hepatic tissues also exhibited necrosis (black arrow), as well as further effects, highlighted by binucleation (blue arrow), under CU or Gem administrations (Figure 12c). Notably, only the Gem injection resulted in megakaryocytes (green arrow) in the spleen tissues (Figure 12d). Remarkably, necrotic cells were almost absent after treatment with Gem and CU4c, while cells with a normal structure were more frequently detected. Consequently, CU4c synergistically enhanced the efficacy of Gem in suppressing tumor proliferation. Furthermore, CU4c reduced the toxicity of Gem and preserved the functionality of the visceral organs in the xenograft mice.

## 3. Discussion

This research further explores the inhibitory effects of HDAC and the anticancer potential of CU4c, assessing both single and combination therapies against human lung cancer cells through in vitro and in vivo evaluations. The in silico results suggest that CU4c may interact with various amino acids in the catalytic area of human HDAC class I and class II and chelate with zinc ions, HDAC cofactors (Figure 3). Furthermore, CU4c demonstrated the lowest values for binding energy (ΔG) and inhibitory constant (Ki) among all Class I and II HDACs, with binding energies ranging from −5.89 to −7.92 kcal/mol. HDAC VII showed the highest binding energy of −7.92 kcal/mol (Table 1). Binding energy levels nearing zero indicate weaker binding, while more negative values imply stronger binding. Previously, the binding energies of belinostat with HDACs I, II, III, IV, VI, and VII ranged from −8.30 to −9.20 kcal/mol, while for valproic acid, the range was −4.70 to −6.00 kcal/mol [29]. CU4c outperformed some inhibitors, including valproic acid. The most favorable binding energy for CU4c, −7.92 kcal/mol, was observed with HDAC VII. Although this value is lower than the binding energies of belinostat (−8.60 kcal/mol), it is still higher than that of valproic acid (−4.70 kcal/mol) [29]. Moreover, this study demonstrated that CU4c exhibited in vitro HDAC inhibitory activity with IC_50_ of 18.75 ± 0.23 µM (Figure 2a). CU inhibited HDAC activity with an IC_50_ of 115 µM, according to Tatar et al. [30]. Therefore, CU4c showed a greater HDAC inhibitory activity than the CU prototype. CU4c also induced the hyperacetylation of histone H3 in A549 cells (Figure 2b,c). MTT results revealed that CU4c demonstrated significant antiproliferative activity against A549 cells in a dose- and time-dependent manner (IC_50_ = 93.90 ± 1.84, 60.43 ± 2.18, and 40.34 ± 1.20 µM at 24, 48, and 72 h, respectively) (Figure 4a). This effect was notably less prominent in Vero cells, with IC_50_ over 200 µM at all exposure times (Figure 4b). Additionally, SI values exceeded 2, demonstrating specific cytotoxicity toward cancer cells (Table 2). These data suggest that CU4c preferentially eliminated malignant cells while causing fewer negative effects in non-cancerous cells. Consistently, our previous study demonstrated that modified curcumin compounds (CU4g and CU5j) inhibited the growth of HeLa, HCT116, and MCF-7 cells while leaving non-cancerous Vero cells unaffected [26]. Additionally, the curcumin derivative ZYX02-Na suppressed the proliferation of lung cancer A549 cells, exhibiting an IC_50_ value of 71.025 μM after 24 h [31]. The IC_50_ values of CU4c were close to the IC_50_ values of ZYX02-Na against A549 cells. Consequently, CU4c is a potential anticancer drug for lung cancer cells.

Drug resistance negatively impacts cancer treatments. Hence, multiple medications are frequently administered concurrently as a combinatorial therapy. Synergistic medicine combinations inhibited the growth of cancer cells at lower dosages, increasing patient outcomes and lowering adverse effects [32]. Previously, CDDP and TSA (HDAC inhibitor) showed synergistic antitumor effects in lung cancer cells [33]. Similarly, the HDAC inhibitor Vorinostat promoted CDDP’s anticancer activity in NSCLCs [34]. Consistently, the HDAC inhibitor curcumin derivative CU17 enhanced Gem’s activity and reduced its toxicity in lung cancer cells (in vitro) and in mouse models [35]. In this study, the combinations of CU4c and CDDP (IC_20_ and IC_30_) exhibited synergistic effects at 24 h exposure (CI = 0.38 and 0.53, respectively). The synergistic impact reduced CDDP dosage by 2.01 to 3.13-fold (Table 3). In addition, the current study showed that CU4c and Gem (IC_20_) synergized at 48 and 72 h (CI = 0.26 and 0.31, respectively). The combination with Gem at IC_30_ showed considerable synergism and nearly additive effects (CI = 0.76 and 0.91), leading to reductions in Gem doses from 8.37- to 16.00- and 2.44- to 3.83-fold at 48 and 72 h exposures, respectively (Table 3). Moreover, the combination therapies of CU4c with CDDP or Gem notably inhibited the growth of A549 cells in comparison to the single-drug treatments (Figure 5 and Figure 6). These results strongly suggest that CU4c enhance CDDP and Gem’s anticancer effects against lung cancer A549 cells.

The CU4c combination treatment with either CDDP or Gem significantly inhibited the proliferation of A549 cells via a dual mechanism involving cell cycle arrest and the induction of apoptosis. The dysregulation of the cell cycle is dramatically related to malignant transformation, and specific substances may demonstrate anticancer effects through their influence on the cell cycle [36]. CDDP causes cancer cell death by causing DNA damage, which disrupts DNA replication during the S phase [37]. Gem, a classic antimetabolite, interrupts DNA replication and repair, triggering cell cycle checkpoints. Low gemcitabine concentrations produce S-phase arrest, but large amounts may arrest all cell cycle stages, including G2/M [38]. In this study, CDDP in a single treatment arrested the cell cycle in the S phase and enhanced Sub G1 cell numbers against A549 Cells (Figure 7a,b). However, Gem-treatment activated the cell cycle arrest in A549 cells at the S and G2/M phases (Figure 8a,b). Low doses of CU4c (2.75, 5.66, and 16.82 µM) did not impact the cell cycle progression of A549 cells, but the higher concentration (18.92 µM) induced cell cycle arrest between the S and G2/M phases (Figure 7 and Figure 8). Similarly, CU and CU17 also induced S and G2/M phase arrests in A549 cells [20]. Dai et al. reported that WZ35, a curcumin derivative, caused lung cancer cell arrest at the G2/M phase, disrupting the cell cycle and possibly causing apoptosis in lung cancer cells [39]. Furthermore, the combination of CU and Gem enhanced cellular apoptosis and induced G2/M phase arrest in gemcitabine-resistant cholangiocarcinoma cells [40]. A previous study demonstrated that the combined treatment with Gem and CU17 resulted in S and G2/M phase arrests in A549 cells [35]. Weir et al. reported that CU inhibited the growth of ovarian cancer cells resistant to cisplatin by inducing superoxide production, causing G2/M phase arrest, and promoting apoptosis [41]. Consistently, in this study, all combination therapies induced S and G2/M phase arrest in the A549 lung cancer cells (Figure 7a,b and Figure 8a,b).

p21, an essential cyclin-dependent kinase inhibitor, directly inhibits cyclin-CDK1, cyclin-CDK4/6, and cyclin-CDK2 complexes, arresting the cell cycle [42]. The tumor-suppressor transcription factor p53 affects apoptosis and cell survival by recognizing DNA damage. Previous research suggests that p53 induces p21 expression, arresting the cell cycle [43]. In this study, we showed that CU4c combined with CDDP significantly upregulated both p53 and p21 (Figure 7c,d), corroborating the findings from flow cytometry. Similarly, CU4c and Gem greatly upregulated p21 protein expression relative to other single medication treatments (Figure 8c,d). The results suggest that the combination of CU4c and CDDP induced cell cycle arrest at the S and G2/M phases in A549 lung cancer cells through the upregulation of p53 and p21 protein expression. Furthermore, the combination of CU4c and Gem also induced S and G2/M phase arrests in A549 cells by upregulating p21 protein expression. Overall, p21 is significantly upregulated by p53, leading to p21’s crucial role in cell cycle arrest. Nonetheless, raised p21 levels may also occur through a p53-independent mechanism, such as phosphorylation (e.g., Akt, which aids in retaining p21 in the cytoplasm) or by N-terminal ubiquitination [44,45].

Apoptosis, also referred to as programmed cell death, is regulated by proteins classified as antiapoptotic and proapoptotic. An imbalance between these proteins can lead to tumor formation. The Bcl-2 family, including Bcl-2, Bcl-xL, and Mcl-1, consists of key antiapoptotic proteins linked to lung cancer progression [46]. Our results demonstrate that cotreatment with CU4c and Gem or CDDP significantly enhanced apoptosis in A549 cells when compared to single treatment (Figure 9a,b and Figure 10a,b). Additionally, combining CU4c with Gem or CDDP maintained the levels of the proapoptotic Bax protein while significantly reducing the levels of the antiapoptotic Bcl-2 protein (Figure 9c,d and Figure 10c,d). Statistical analysis showed that the coadministration of CU4c with Gem or CDDP significantly increased the ratio of proapoptotic Bax to antiapoptotic Bcl-2 compared to single-drug treatments (Figure 9e and Figure 10e). Western blot analysis confirmed the inhibition of Bcl-2 and activation of Bax, correlating with flow cytometry results that indicated increased apoptosis. Additionally, the p21 protein enhances apoptosis by upregulating Bax and Bak and downregulating Bcl-2 [44,47]. An increased Bax/Bcl-2 ratio reduces cellular resistance to apoptosis activation, leading to increased cell death and a decreased risk of tumor formation [41]. In summary, our findings suggest that the combination of CU4c with Gem or CDDP enhances cellular apoptosis, increasing the sensitivity of Gem and CDDP in A549 cells through the activation of the Bax/Bcl-2-dependent intrinsic apoptosis pathway.

The ERK pathway is essential for regulating cell growth and differentiation. Oxidative stress can act as a signal that helps cells survive. However, too much oxidative stress can cause serious damage and lead to cell death [48]. Previous research has demonstrated that ERK phosphorylation is important in cisplatin-induced apoptosis in HeLa cells. The cytotoxic effects of cisplatin are related to DNA damage and increased reactive oxygen species [49]. Moreover, ERK activity may promote cytochrome c release by modulating the expression of Bcl-2 family proteins. The MEK/ERK pathway upregulates proapoptotic Bcl-2 family members like Bax and downregulates antiapoptotic members like Bcl-2 and Bcl-Xl [50]. Here, combining CU4c with CDDP or Gem significantly enhances ERK1/2 phosphorylation in A549 cells, suggesting that CU4c may synergistically activate the ERK signaling pathway, increasing the Bax/Bcl-2 ratio (Figure 9c,d and Figure 10c,d). HDAC inhibitors cause decreased tumor cell proliferation, differentiation, and viability without damaging normal tissues. Clinically, lung cancer patients with high HDAC1, HDAC2, and HDAC6 levels have a poor prognosis [51]. Additionally, HDAC inhibitors can increase levels of p21 and enhance the efficacy of carboplatin treatment against lung cancer cells [52]. Our previous study indicated that a combination of CU17 and Gem improved histone H3 hyperacetylation [35]. This study’s findings also indicate that combining CU4c with CDDP or Gem significantly upregulated histone H3 hyperacetylation compared to single-drug treatments (Figure 9c,d, and Figure 10c,d). HDAC inhibitors can halt the cell cycle by upregulating p21 [53], suggesting that increased acetylated histone H3 may enhance apoptosis and inhibit cell cycle progression in response to CU4c and CDDP or Gem treatments.

In this study, combination treatment of CU4c and Gem inhibited lung carcinoma cell proliferation more effectively than combined CU4c and CDDP treatment; therefore, CU4c combination treatment with Gem was chosen for further in vivo testing. Following 21 days of therapy, CU4c alone did not demonstrate a significant decrease in tumor volume, growth, or weight when compared to those treated alone with CU (Figure 11c–e). While CU caused necrosis in renal and hepatic tissues, CU4c did not have any adverse effects on these tissues (Figure 12b,c). Importantly, the combination of CU4c and Gem significantly enhanced tumor growth inhibition (Figure 11c–f). This treatment led to a considerable reduction in the proliferating cell number and produced high-power images showing the apoptotic characteristics in the tumor tissues (Figure 12a). Interestingly, this combined treatment had no notable impact on body weight compared to the group that received only one type of therapy (Table 4). However, mice treated with Gem alone exhibited necrosis in the liver and kidneys, as confirmed by histopathological analysis (Figure 12b,c). Furthermore, intermittent binucleation was detected in the liver tissues of mice treated with Gem or CU (Figure 12c). Binucleation is a result of cellular damage and is a kind of chromosomal hyperplasia, often seen in regenerated cells [54]. Gem monotherapy led to megakaryocyte formation, resulting in necrosis of splenic tissues (Figure 12d), which correlated with a rise in spleen weights suggestive of splenomegaly (enlarged spleen) (Table 4). Megakaryocytes, typically located in bone marrow, are responsible for platelet production. Their presence and elevated numbers in the spleen may lead to splenomegaly in specific conditions, notably myelofibrosis and other hematological disorders [55]. Previous research indicated that liver and kidney sections from Gem-treated animals showed signs of hepatic tissue breakdown and necrotic regions [56,57,58]. Nonetheless, Gem combination therapies have shown promise in reducing these adverse effects. For instance, LPE has been shown to greatly reduce liver damage caused by Gem therapy [59,60]. Our earlier findings demonstrated that administering Gem together with CU17 resulted in a more uniform cellular architecture in liver, kidney, and spleen tissues, accompanied by reduced tissue damage [35]. The results from this study indicated that necrotic cells were nearly absent after treatment with Gem and CU4c, while cells with normal structure were observed more frequently (Figure 12b–d). Thus, the concurrent administration of CU4c and Gem led to diminished tumor proliferation and reduced toxicity to the liver and kidneys. Our findings suggest that the combination of Gem and CU4c has the potential to enhance therapeutic effectiveness while minimizing toxicity.

## 4. Materials and Methods

### 4.1. Materials

The powdered turmeric rhizome was bought from a natural pharmacy in Khon Kaen province, Thailand. RPMI-1640 medium, penicillin/streptomycin, and trypsin-EDTA were purchased from Thermo Fisher Scientific Inc. (Grand Island, NY, USA). The fetal bovine serum (FBS) was obtained from Cytiva (Kremplstrasse, Pasching, Austria). Both Annexin V-FITC and 3-(4,5-dimethylthiazol-2-yl)-2,5-diphenyltetrazolium bromide (MTT) were obtained from Biolegend (San Diego, CA, USA) and Invitrogen (Eugene, OR, USA), respectively. The source of gemcitabine hydrochloride (Gem) (Purity of ≥98%), polyethylene glycol (PEG) 400, olive oil, and propidium iodide (PI) (Purity of ≥94%) was Sigma-Aldrich Corporation (St. Louis, MO, USA). The source of cisplatin (CDDP) (Purity of ≥99.9%) was the European Pharmacopoeia (Allée Kastner, Strasbourg, France). Additionally, the primary antibodies (anti-p53, anti-Bcl-2, anti-Bax, anti-Ac-H3, anti-pERK1/2, anti-p21, anti-Total ERK1/2, and anti-β-actin) and secondary antibodies (anti-mouse IgG and anti-rabbit IgG conjugated to horseradish peroxidase) were supplied by Cell Signaling (Beverly, MA, USA). All chemicals and reagents are graded for research use (Research grade).

### 4.2. Cell Lines and Culture Conditions

The A549 cell line (human lung adenocarcinoma) was graciously supplied by Dr. Arunporn Itharat, Thammasat University, Bangkok, Thailand. The noncancerous Vero cell line was generously donated by Dr. Sahapat Barusrux, Walailak University, Nakhon Si Thammarat, Thailand. The cells were cultured in RPMI-1640 media. Additionally, 10% FBS, 100 U/mL penicillin, and 100 µg/mL streptomycin were mixed into the medium to enhance cell culture growth. The cells were grown at 37 °C in a humidified environment with 5% CO_2_.

### 4.3. The Extraction and Isolation of Curcumin

For the synthesis of CU4c, curcumin was generated by extracting and isolating turmeric root, as previously described [61]. Briefly, the powdered turmeric was extracted three times using dichloromethane (CH_2_Cl_2_) (1000 mL each extraction). The crude extract was then condensed in a rotary evaporator at low pressure to produce a dichloromethane extract. Silica gel column chromatography was used to isolate the dichloromethane extract. The extract was then eluted using methanol (MeOH), hexane, and ethyl acetate (EtOAc). Following collection, the elution solutions were separated into four fractions: DT1, DT2, DT3, and DT4. Three subfractions (DT2-1 to DT2-3) were obtained from the DT2 fraction using a hexane–EtOAc gradient (from 10:0 to 5:5) elution and flash silica gel column chromatography. Following that, DT2-1 and DT2-2 were identified using thin layer chromatography (TLC) with a purifying hexane–EtOAC (9:1) mobile phase. The reaction was monitored using TLC under UV light (λ = 254 nm), then stained with an anisaldehyde solution for identification via Rf values and colorimetric analysis. The end products were solid yellow demethoxycurcumin and dihydrocurcumin. Curcumin was obtained as a solid orange color by separating DT2-3 using silica gel column chromatography and a CH_2_Cl_2_-MeOH (9:1) mobile phase. The NMR results for all products were similar to those documented by Venkateswarlu [62]. The purities based on HPLC chromatograms of CU4c and CU were provided in the Appendix A (Appendix A, respectively).

### 4.4. Curcumin Derivative CU4c Synthesis

The synthesis of CU4c was conducted following the prior methodology [26]. In summary, curcumin (1.016 g: 2.7580 mmol) was solubilized in acetone (50 mL). Subsequently, potassium carbonate (5.7918 mmol) was added to the curcumin solution and stirred for 10 min at room temperature. Following that, diethyl sulfate (5.7918 mmol) was added to the solution. The reaction was refluxed and monitored using TLC for approximately 3 to 6 h. A combined solution was rinsed with water (50 mL) and subjected to ethyl acetate extraction three times (50 mL each extraction). The combined organic phase was washed with brine and dried with anhydrous sodium sulfate, and the vehicle was removed under vacuum pressure, yielding crude products in brown oil. Purification was achieved using column chromatography with a gradient of 20–80% ethyl acetate/hexane, resulting in the isolation of (1E,4Z,6E)-1,7-bis(4-ethoxy-3-methoxyphenyl)-5-hydroxyhepta-1,4,6-trien-3-one (CU4c) (Figure 1), an orange powder, and melting point 130–133 °C. The NMR spectra and HPLC chromatograms of CU4c were provided in Appendix A. The compounds’ NMR spectra matched the previously reported data [26].

### 4.5. In Vitro HDAC Inhibitory Activity Assay

The Fluor-de-Lys HDAC Fluorometric Activity Assay Kit measured HDAC activity inhibition in vitro (Biomol, Enzo Life Sciences International, Inc., New York, NY, USA). A 96-well microplate was added with the HeLa nuclear extract, buffer, and CU4c. The plate was then incubated for 5 min at 37 °C. Following that, each well was combined with a substrate and incubated at 37 °C for 15 min. The addition of a developer suppressed the reaction. The plate was then incubated at room temperature for 15 min. Samples were analyzed with a Thermo Scientific Varioskan Flash Spectral Scanning Multimode Reader (Thermo Fisher Scientific, Grand Island, NY, USA) with excitation at 360 nm and emission at 460 nm. Trichostatin A (TSA) served as a positive control. A reduction in the fluorescence signal indicated the suppression of HDAC activity. All tests were conducted in triplicate.

### 4.6. Molecular Docking of CU4c to HDAC Crystal Structures

The CU4c–HDAC interaction was studied using AutoDock4 (V4.2.6). The Protein Data Bank (http://www.rcsb.org) (accessed on 25 November 2024) provided the crystal structures of HDACs 1–4, and HDACs 6–8 [PDB entry codes: 4BKX, 3MAX, 4A69, 2VQW, 6UO2, 3C0Z and 1T64, respectively] for molecular docking investigations. All water, noninteracting ions, and ligands were removed. The ADT program added all missing hydrogen and side-chain atoms. System Gasteiger charges were estimated. The molecular modeling application GaussView 3.0 was used to construct ligands. Gaussian 03W optimized AM1-level ligands. A 60 × 60 × 60 grid box with 0.375 Å spacing was used to cover the majority of the HDAC protein surface.

### 4.7. Cell Viability Assay

The MTT assay was utilized to evaluate the effect of CU4c on cell proliferation in A549 and Vero cells. A total of 8 × 10^3^ cells per well were seeded into a 96-well plate and incubated at 37 °C for 24 h. The cells were then treated with various concentrations of CU4c for 24, 48, and 72 h. In addition, combination treatments using CU4c at different concentrations and subtoxic concentrations of CDDP (IC_20_ = 21.00, 7.12, and 1.69 µM; IC_30_ = 32.73, 9.16, and 2.68 µM for 24, 48, and 72 h, respectively) or Gem (IC_20_ = 0.68 and 0.35 µM; IC_30_ = 1.30 and 0.55 µM for 48 and 72 h, respectively) were also performed at these time points. After treatment, the cells were incubated with fresh medium containing MTT for 3 h at 37 °C. Following this incubation, the medium was removed, and the formazan crystals were dissolved in DMSO. The absorbance of the formazan solution was measured at 570 nm using an iMark™ Microplate Absorbance Reader (Bio-Rad, Hercules, CA, USA), with a reference wavelength of 655 nm. The following equation was used to determine the percentage of cell viability.$$\mathrm{Cell}\;\mathrm{viability}\;(\%)=\frac{A_{570\;\mathrm{treatment}}-{O.D.}_{655\;\mathrm{treatment}}}{A_{570\;\mathrm{vehicle}\;\mathrm{control}}-{O.D.}_{655\;\mathrm{vehicle}\;\mathrm{control}}}\;\times \;100$$
where O.D. and A are the optical density and absorbance, respectively.

### 4.8. Selectivity Index (SI)

The selectivity of the compounds was quantified by their SI value, where an SI value more than 2 indicates preferential toxicity towards cancer cells, while an SI value less than 2 signifies overall toxicity towards both cancerous and normal cells [27]. The SI value was determined using the equation:(1)$$\mathrm{SI}=\frac{{\mathrm{IC}}_{50\;\mathrm{noncancer}\;\mathrm{cells}}\;}{{\mathrm{IC}}_{50\;\mathrm{cancer}\;\mathrm{cells}}}$$

### 4.9. Drug Interaction

The combination index (CI) theorem of Chou and Talalay [28] was used to evaluate the type of drug interaction (synergistic, additive, or antagonistic effects) between CU4c and CDDP or Gem. The equation of CI was calculated by the following equation:(2)$$\mathrm{CI}=\frac{D1}{\mathrm{Dx}1}+\frac{D2}{\mathrm{Dx}2}+\alpha \frac{D1D2}{\mathrm{Dx}1\mathrm{Dx}2}$$
where D1 is the dose of CDDP or Gem in combination with CU4c that induces 50% cell viability; D2 is the dose of CU4c administered in conjunction with CDDP or Gem to induce 50% cell viability; Dx1 is the dose of CDDP or Gem alone to induce 50% cell viability; Dx2 is the dose of CU4c alone to induce 50% cell viability; and α = 1 signifies mutually nonexclusive modes of drug action. CI > 1.10 signifies antagonism; CI = 0.90–1.10 signifies additive effect; and CI < 0.90 signifies synergistic effect.

The fold reduction in the dose of combination therapy was illustrated through the dose reduction index (DRI), which provides the effect level in comparison to the dose of a single agent. The DRI was determined by employing the following equation:$$\mathrm{DRI}=\frac{\mathrm{Dx}}{D}$$
where D represents the dosage of a drug combined with other drugs to achieve 50% cell viability, whereas Dx denotes the dose of a single drug required to attain 50% cell viability.

### 4.10. Cell Cycle Analysis

The distribution of the cell cycle was analyzed using a PI staining assay. A549 cells were cultured for 24 h at a density of 1 × 10^6^ cells per dish in 60 × 15 mm^2^ plates. Afterward, the cells were treated with CU4c, CDDP, and Gem, either as a single drug or in combination, under synergistic conditions. For the combination treatment with CDDP, A549 cells were exposed to the following: vehicle control (0.5% ethanol + 0.5% DMSO), CDDP (21.00 and 32.73 µM), CU4c (2.78 and 5.66 µM), CDDP (21.00 µM) + CU4c (5.66 µM), and CDDP (32.73 µM) + CU4c (2.78 µM) for 24 h. For the Gem combination treatment, A549 cells received a vehicle control (0.5% ethanol + 0.5% DMSO), Gem (0.68 and 1.30 µM), CU4c (16.82 and 18.92 µM), Gem (0.68 µM) + CU4c (18.92 µM), and Gem (1.30 µM) + CU4c (16.82 µM) for 48 h. Following the incubation, the cells were harvested, washed and then permeabilized as described previously [20]. After rinsing with 1X PBS and incubating with RNase A, the cells were stained in the dark with PI at room temperature for 45 min. The stained cells were quantified using a BD FACSCanto II flow cytometer (Becton Dickinson, San Jose, CA, USA), and the BD FACSDiva software v9.2 was used to analyze the percentages of the different cell cycle phases.

### 4.11. Cellular Apoptosis Detection

The Annexin V-FITC and PI staining technique measured apoptotic induction. A549 cells were plated at 1 × 10^6^ cells/dish on 60 × 15 mm^2^ dish plates for 24 h. After that, cells were treated as in Section 4.10. After treatment, cells were trypsinized, centrifuged, and washed as described previously [20]. Resuspended cells in 1X Annexin V-binding buffer were stained with Annexin V-FITC/PI for 15 min at room temperature in the dark. After staining, cells were gently mixed with 1X Annexin V-binding buffer. The BD FACSCanto II flow cytometer (Becton Dickinson, San Jose, CA, USA) and BD FACSDiva software measured and analyzed apoptotic cells.

### 4.12. Western Blot Analysis

Western blotting was conducted to investigate key proteins associated with cell cycle progression and apoptotic pathways (both proapoptotic and antiapoptotic proteins). A549 cells were cultivated at a density of 1 × 10^6^ cells per 60 × 15 mm^2^ dish for 24 h, after which the cells were treated as previously described for the cell cycle progression. Following incubation, cellular protein was extracted using a RIPA buffer supplemented with protease inhibitor cocktail (Amresco Inc., Solon, OH, USA), incubated on ice for 20 min. The protein content in the samples was quantified using a Bio-Rad protein assay kit based on the Bradford technique (Bio-Rad Laboratories, Hercules, CA, USA). Total protein was separated using sodium dodecyl sulfate polyacrylamide gel electrophoresis (SDS-PAGE). Subsequently, the proteins were transferred to a polyvinylidene fluoride (PVDF) membrane. The blotted membranes were blocked with a 3% nonfat milk solution for 1 h at room temperature. The blotted membrane was incubated overnight at 4 °C with the following primary antibodies: anti-Ac-H3 (#9649, diluted 1:1000), anti-β-actin (#3700, diluted 1:2000), anti-Bax (#2772, diluted 1:1000), anti-Bcl-2 (#4223, diluted 1:1000), anti-pERK1/2 (#9102, diluted 1:1000), anti-total ERK1/2 (#9107, diluted 1:2000), anti-p21 (#2946, diluted 1:1000), and anti-p53 (#2524, diluted 1:1000) (Cell Signaling, Beverly, MA, USA). The blotted membranes were washed in 1X PBST for 2 min twice and then incubated with HRP-conjugated goat anti-mouse (#7076, diluted 1:2000) or anti-rabbit (7074, diluted 1:2000) secondary antibodies at room temperature for 2 h. Then, 1X PBST and PBS washes, each lasting 2 min, were performed twice. The blot was ultimately analyzed using enhanced chemiluminescence reagents (Bio-Rad, Hercules, CA, USA) and X-ray films to examine the immunoreactive bands. The assessment of relative intensity was conducted using the ImageJ software, with β-actin serving as a loading control to normalize protein levels and total ERK1/2 acting as a loading control for pERK1/2.

### 4.13. In Vivo Antitumor Study

After in vitro testing, animal studies are needed to evaluate the drug’s effects and potency in living organisms. Mice are effective animal models for cancer cell transplantation, allowing for clear observation of tumor mass variations for analysis. Female nude mice (BALB/cAJcl-Nu/Nu, 4–6 weeks old) obtained from Nomura Siam International (Bangkok, Thailand) were maintained in individual ventilation cages (IVCs) at Khon Kaen University’s Northeast Laboratory Animal Center. The sample size was calculated based on a two-independent mean comparison (independent *t*-test) [63]. The mice were provided with a regular chow pellet. The standard environments were 20–24 °C, 30–60% relative humidity, and illumination at 350–400 Lux, following a 12/12 h light/dark cycle. The Institutional Animal Care and Use Committee at Khon Kaen University gave its approval for the animal tests (approval number IACUC-KKU-68/67; registration date 25 July 2024). The researchers also followed the National Research Council of Thailand’s Ethical Principles and Guidelines for the Use of Animals in Scientific Purposes. The research complied with ARRIVE guidelines. A549 cells (6 × 10^6^) were subcutaneously implanted into the right axilla of mice to create lung cancer xenografts using 0.1 mL of serum-free media and Matrigel (Corning, Tewksbury, MA, USA). Five groups were randomly assigned once tumor development reached 100 mm^3^ (n = 4–5). Group 1 received 20% PEG400 in olive oil as a vehicle control. Group 2 received 30 mg/kg CU and Group 3 received 30 mg/kg CU4c. Group 4 received 50 mg/kg Gem, whereas group 5 received 50 mg/kg Gem and 30 mg/kg CU4c. The treatments were divided into two sets. In the first set, mice from groups 1 to 3 received treatment, while in the second set, mice from groups 4 and 5 were treated. The mice received intraperitoneal administrations of treatment, either alone or in combination, with a maximum volume of 10 mL/kg, and tumor volume and body weight were monitored every three days for 21 days. A veterinarian conducted daily health assessments to ensure the well-being of the mice. Dropout criteria included severe infections or mortality during the study period. On the last day, following CO_2_ anesthesia, tumors and visceral organs (liver, kidney, and spleen) of mice were excised and weighed. The following equation represents tumor inhibition using the percentage inhibition ratio. Tumor volume was (mm^3^) = (a × b^2^)/2, where a represents length and b represents breadth in mm. The mice were slaughtered, and the tumors and visceral organs (kidneys, liver, and spleen) were weighed after the experiment. The relative tumor volume (RTV), tumor growth inhibition ratio (%TGI), and body weight change (BWC) were calculated according to the previous publication [60].

### 4.14. Histological Examination

The tumors, livers, kidneys, and spleens were fixed with 10% formalin. Following this, the tissues were subjected to dehydration, tissue-processing, and paraffin-embedding. The samples were meticulously placed on a glass slide after being sliced to a thickness of 4 µm using a microtome. The tissue sections were deparaffinized in xylene, rehydrated in ethanol (99, 95, and 70%), and finally washed with deionized water. Hematoxylin and eosin were used to stain the rehydrated tissue sections. Subsequently, the representative regions were captured at 400 magnification using an Olympus BX60 Fluorescence Microscope (Olympus Corporation, Tokyo, Japan).

### 4.15. Statistical Analysis

Statistical analyses were conducted using SPSS 22.0 (IBM, Manassas, VA, USA). The results are shown as mean ± standard deviation (SD) and mean ± standard error of the mean (SEM) for independent experiments. Graphical representations were created using GraphPad Prism 8.0 (GraphPad Software, La Jolla, CA, USA). A one-way analysis of variance (ANOVA) was used to evaluate statistically significant differences between the control and experimental groups, followed by Duncan’s post hoc test for pairwise comparisons. *p* < 0.01 or 0.05 was deemed statistically significant. All tests were conducted in triplicate.

## 5. Conclusions

CU4c possessed significant HDAC-inhibitory and antiproliferative effects on A549 cells (IC_50_ = 93.90 ± 1.84, 60.43 ± 2.18, and 40.34 ± 1.20 µM at 24, 48, and 72 h, respectively) while showing less toxicity towards noncancerous Vero cells (IC_50_ > 200 µM at 24, 48, and 72 h). Moreover, the combined administration of CU4c and CDDP enhanced cellular sensitivity by arresting the S and G2/M phases of the cell cycle and triggering apoptosis in lung cancer cells through the upregulation of p21, p53, pERK1/2, AcH3, and the Bax/Bcl-2 expression ratio. Furthermore, the synergistic impact of Gem and CU4c significantly improved cell cycle arrest during the S and G2/M phases and promoted apoptosis in the human lung cancer cell line via the upregulation of p21, pERK1/2, AcH3, and the Bax/Bcl-2 expression ratio. In vivo experiments in xenograft mice confirmed that the combination of CU4c and Gem substantially suppressed tumor development while causing limited damage to visceral organs. Overall, CU4c may serve as a promising alternative therapeutic agent for lung cancer treatment. Additional research in large-scale animal models is necessary to validate this combination regimen as an effective treatment for NSCLC. Our results support its clinical implementation in patients with NSCLC to enhance therapeutic response to gemcitabine-based regimens.

## Figures and Tables

**Figure 1 pharmaceuticals-18-00960-f001:**
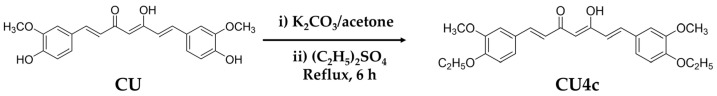
Synthesis of curcumin derivative CU4c (CU4c).

**Figure 2 pharmaceuticals-18-00960-f002:**
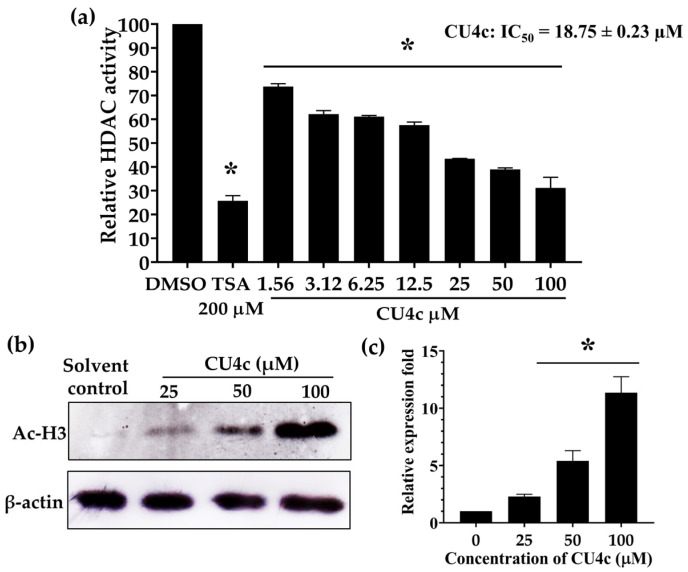
HDAC-inhibitory activity of CU4c. (**a**) The Fluor-de-Lys HDAC Fluorometric Activity Assay was used to determine HDAC-inhibitory activity of CU4c in vitro. The relative HDAC activities were presented by bar graphs concerning the control (DMSO). TSA was used as a positive control. Each value represents the mean ± SEM from three independent experiments. (**b**) Western blot analysis of histone H3 hyperacetylation in A549 cells after exposure to varying concentrations of CU4c for 24 h. Vehicle treatment consisting of 0.5% ethanol and 0.5% DMSO (%*v*/*v*) was used as a negative control. β-Actin was used as a loading control for Western blot analysis. (**c**) The bar graphs represented relative folds of protein expression, calculated using the intensity of the protein band in comparison to a loading control. Data are expressed as mean ± SEM of three independent experiments conducted in triplicate. * *p* < 0.05 indicates a significant difference between the treatment and vehicle control.

**Figure 3 pharmaceuticals-18-00960-f003:**
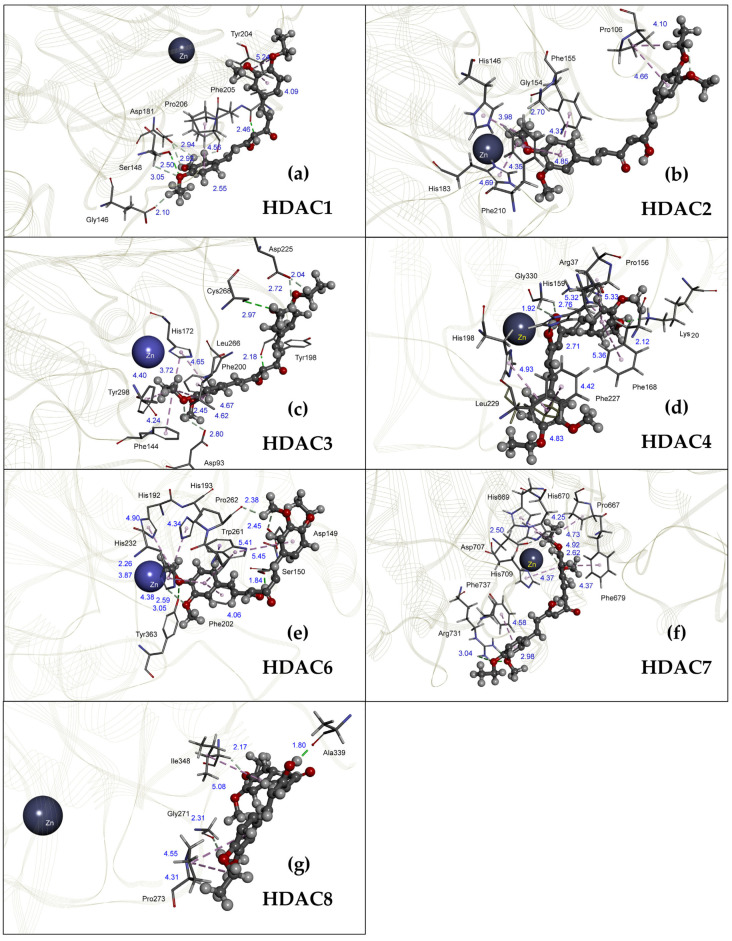
The engagement of CU4c with the active sites of (**a**) HDAC1, (**b**) HDAC2, (**c**) HDAC3, (**d**) HDAC4, (**e**) HDAC6, (**f**) HDAC7, and (**g**) HDAC8. Black = Carbon atom, Grey = Hydrogen atom, Red = Oxygen atom, and Blue = Nitrogen atom.

**Figure 4 pharmaceuticals-18-00960-f004:**
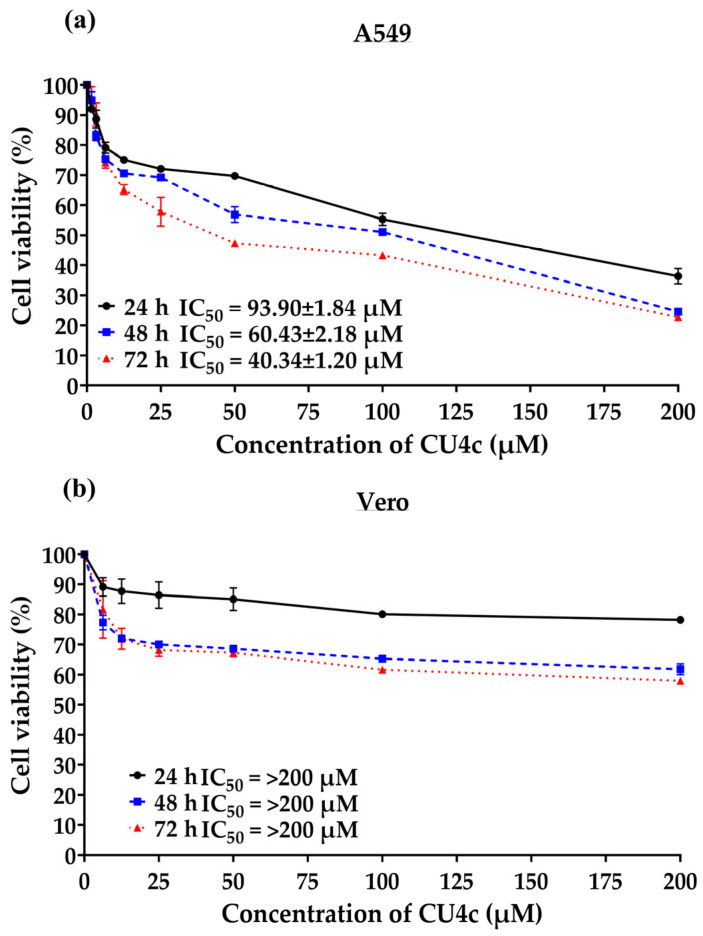
Antiproliferative activity of CU4c. The impact of CU4c on the proliferation of (**a**) A549 and (**b**) noncancerous Vero cells was evaluated after treatment for 24, 48, and 72 h. The antiproliferative effects were assessed using the MTT assay. The data are presented as the percentage of cell viability relative to the vehicle control (0.50% ethanol + 0.50% DMSO), designated as 100% viability. The IC_50_ values are reported as the mean ± SEM from three independent trials.

**Figure 5 pharmaceuticals-18-00960-f005:**
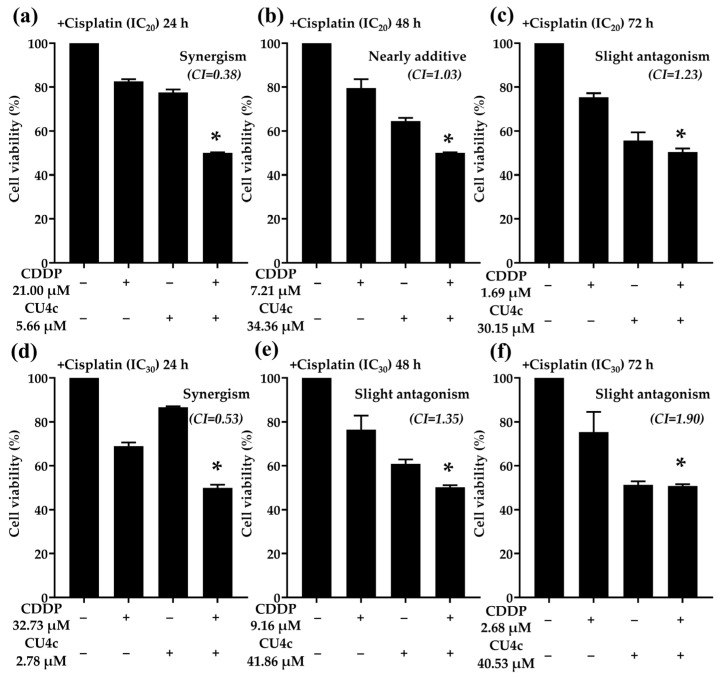
Antiproliferative effects of the combined treatments of CU4c and CDDP against A549 cells at 24, 48, and 72 h. A549 cells were administered CU4c at an IC_50_ dosage in conjunction with CDDP at subtoxic doses of IC_20_ (**a**–**c**) and IC_30_ (**d**–**f**) for 24, 48, and 72 h exposures, respectively. Cell viability was measured as a percentage relative to the control group, which consisted of cells treated with the vehicle (0.50% ethanol + 0.50% DMSO). A statistically significant reduction in cell viability, indicated by * *p* < 0.01, was observed when compared to the monotherapy with CDDP.

**Figure 6 pharmaceuticals-18-00960-f006:**
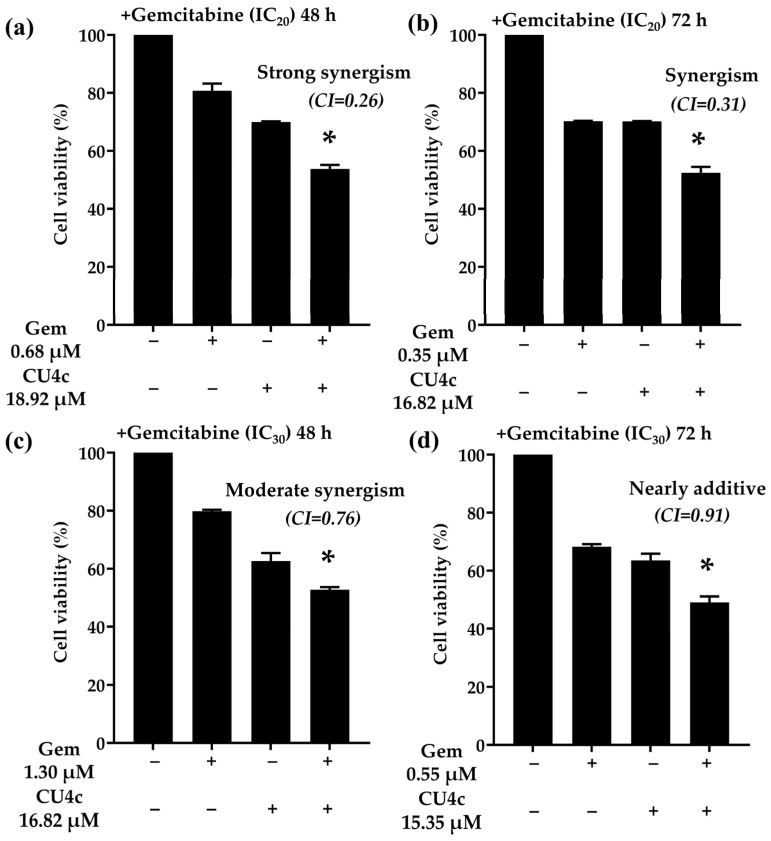
Antiproliferative effects of the co-treatments of CU4c and Gem against A549 cells at 48 and 72 h. A549 cells were treated with CU4c at an IC_50_ dosage along with Gem at subtoxic doses of IC_20_ (**a**,**b**) and IC_30_ (**c**,**d**) for 48 and 72 h exposures, respectively. Cell viability was measured as a percentage relative to a vehicle control treatment. “*” denotes a statistically significant reduction (*p* < 0.01) compared to Gem monotherapy.

**Figure 7 pharmaceuticals-18-00960-f007:**
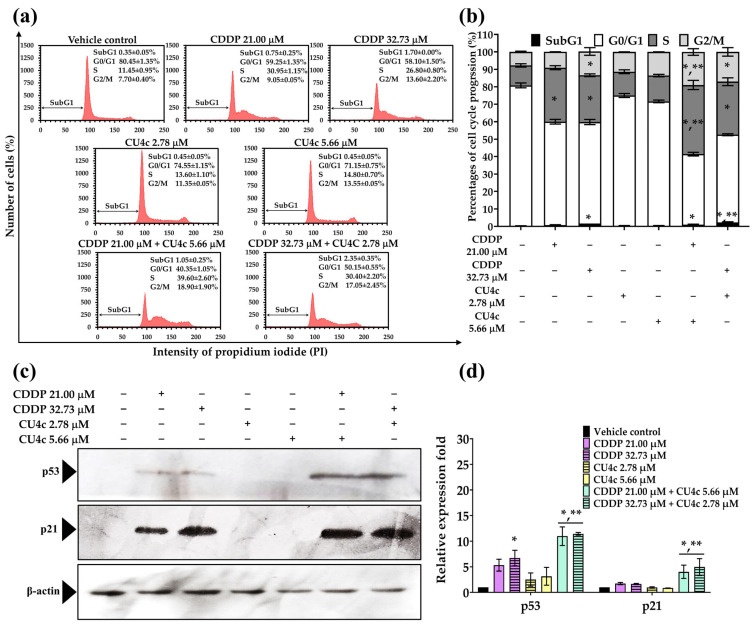
Effect of CU4c and CDDP in both single and combination treatments on cell cycle distribution in A549 cells. The cells were treated with CDDP at concentrations of 21.00 and 32.73 µM, as well as CU4c at 5.66 and 2.78 µM, both individually and in combination, for 24 h. (**a**) Representative DNA histograms illustrate the cell cycle distribution of A549 cells following the treatments. (**b**) Bar graphs display the mean percentages of cells in each phase of the cell cycle. (**c**) The levels of protein expression for p21 and p53 and (**d**) their relative expression folds are shown, using β-actin as a loading control. Data are presented as mean ± SEM (n = 3), with * *p* < 0.05 indicating significance compared to the control, and ** *p* < 0.05 indicating significance compared to the single treatment.

**Figure 8 pharmaceuticals-18-00960-f008:**
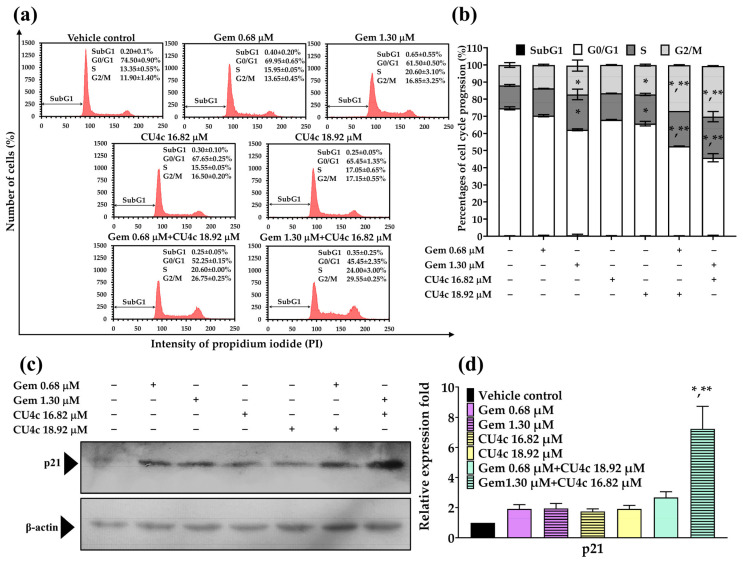
The impact of CU4c and Gem on cell cycle progression in A549 cells was analyzed with both single and combination therapies. The cells were treated with Gem at doses of 0.68 µM and 1.30 µM, and with CU4c at doses of 16.82 µM and 18.92 µM, either individually or in combination, for 48 h. (**a**) Representative DNA histograms demonstrate the cell cycle distribution of A549 cells after the treatments. (**b**) Bar graphs illustrate the average percentages of cells in each cell cycle phase. (**c**) The protein expression levels of p21 and (**d**) their corresponding relative expression folds are shown, using β-actin as a loading control. * *p* < 0.05 denotes significance relative to the control, and ** *p* < 0.05 indicates significance relative to the single treatment.

**Figure 9 pharmaceuticals-18-00960-f009:**
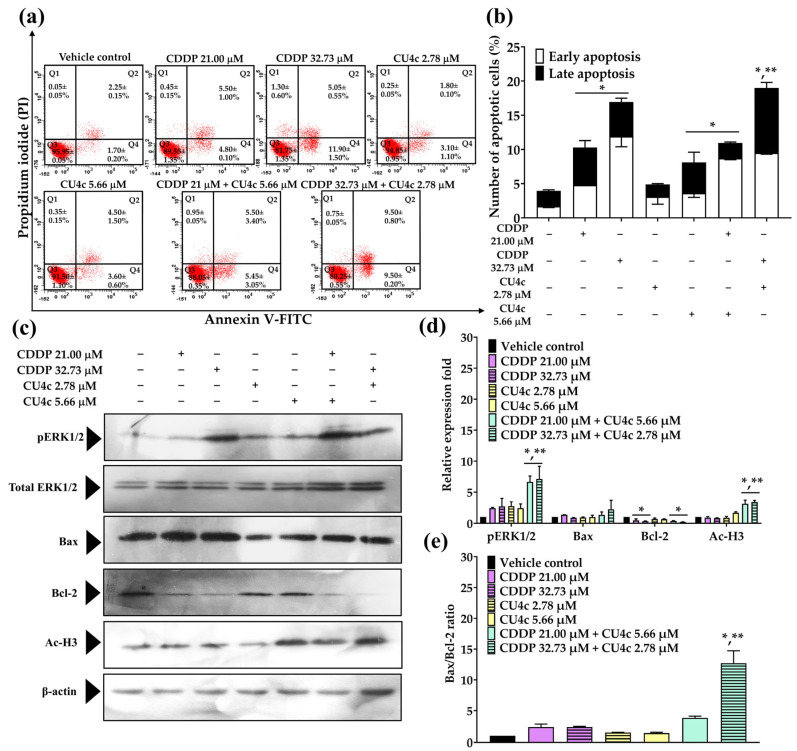
The impact of single and combined drug treatments of CU4c and CDDP on the induction of apoptosis in A549 lung cancer cells. Cells were exposed to CU4c and/or CDDP for a duration of 24 h. Apoptotic cells were detected with the Annexin V-FITC/PI assay. (**a**) Dot plots illustrate the typical Annexin V-FITC/PI data from three consistent experiments. (**b**) The proportion of apoptotic cells after the specified therapy is shown in a bar graph. pERK1/2, Ac-H3, Bax, and Bcl-2 expression levels in A549 cells after various therapies were evaluated by Western blot analysis. (**c**,**d**) The optical densities of protein bands were measured using ImageJ 1.52a. (**e**) The ratio of Bax/Bcl-2 expression was reported, with β-actin serving as a loading control. All data are shown as mean ± SEM (n = 3, * *p* < 0.05 compared to control and ** *p* < 0.05 compared to single treatment.

**Figure 10 pharmaceuticals-18-00960-f010:**
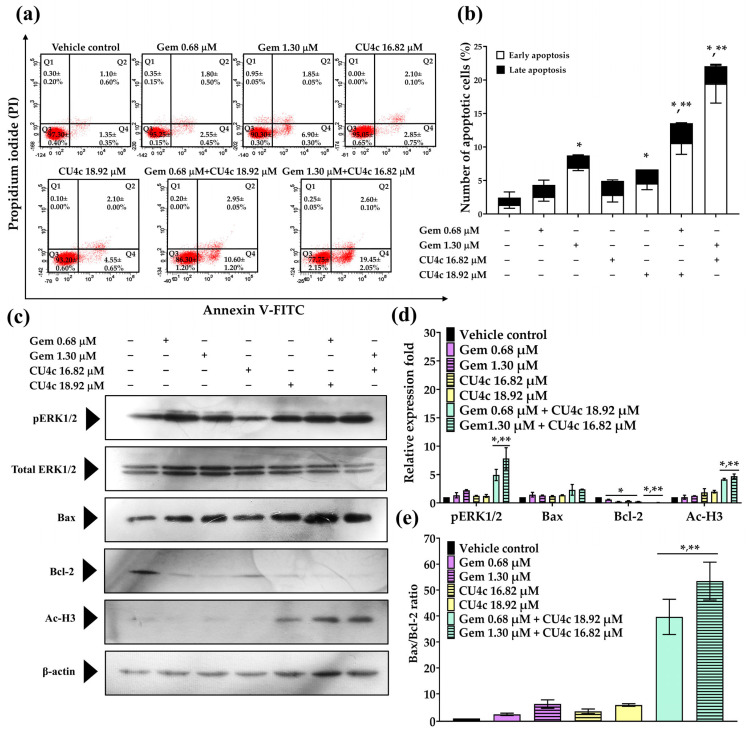
The effect of CU4c and Gem on the induction of apoptosis in A549 lung cancer cells was studied through both single and combined drug treatments. The cells were treated with CU4c and/or Gem for 48 h. The Annexin V-FITC/PI assay was used to identify apoptotic cells. (**a**) Dot plots represent typical Annexin V-FITC/PI data from three consistent experiments. (**b**) A bar graph illustrates the proportion of apoptotic cells following the specified treatments. pERK1/2, Ac-H3, Bax, and Bcl-2 expression levels in A549 cells were assessed after administering various therapies using Western blotting. (**c**,**d**) ImageJ software was used to quantify the optical densities of these proteins. (**e**) The Bax/Bcl-2 expression ratio was calculated, with β-actin and total ERK1/2 serving as the loading controls. The mean ± SEM was reported for all data (n = 3, * *p* < 0.05 compared to a vehicle control, and ** *p* < 0.05 compared to single drug treatments).

**Figure 11 pharmaceuticals-18-00960-f011:**
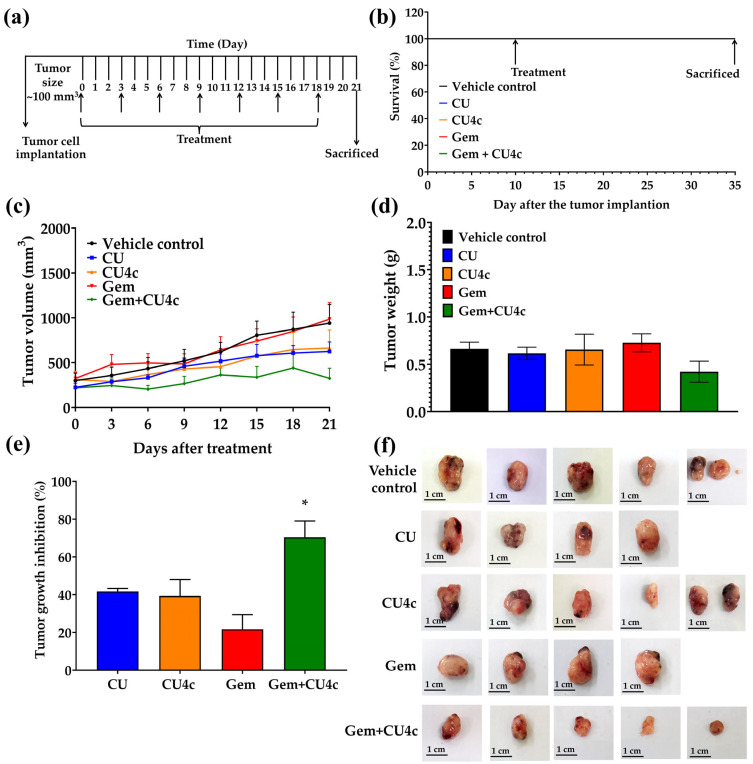
The effects of CU, CU4c, and Gem in A549 xenograft mouse models. (**a**) The experimental plan involved administering CU at 30 mg/kg, CU4c at 30 mg/mL, Gem at 50 mg/kg, and a combination treatment of Gem and CU4c (vehicle control (n = 5), CU (n = 4), CU4c (n = 5), Gem (n = 4), and Gem + CU4c-15 (n = 5). Day 0 is before intervention, with the first injection given on the same day. Treatments and data collection were performed every three days, in which data were collected before each intervention. (**b**) After 21 days of treatment, the survival rates of all the xenograft mice were measured. The arrows in the study indicate the start and end points of the treatment period. (**c**) Tumor volume, (**d**) tumor weight, and (**e**) tumor growth inhibition (%) after administration of the specified drugs were assessed. (**f**) At the end of the observation period, the tumor xenografts were surgically removed. Statistical significance was noted with * *p* < 0.05 when comparing the results with single treatments.

**Figure 12 pharmaceuticals-18-00960-f012:**
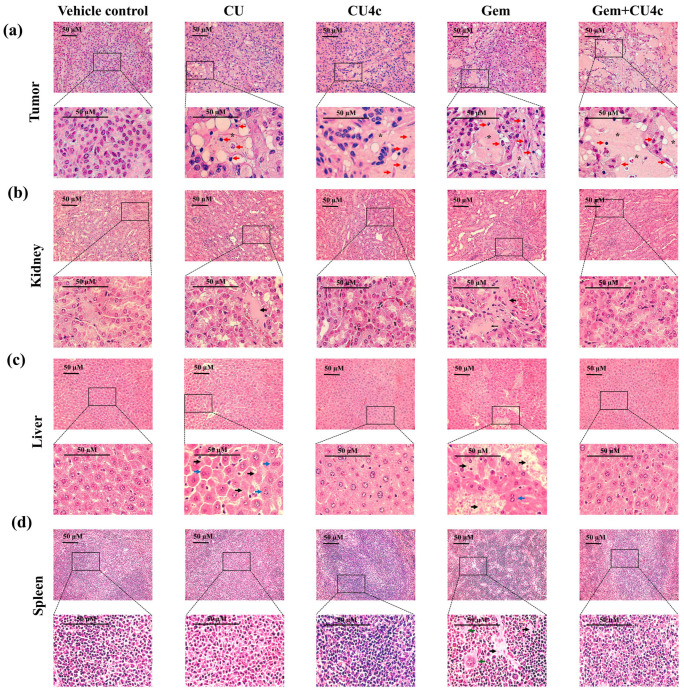
A histopathological examination of mice organs following therapy was conducted. The tissues analyzed included (**a**) tumor, (**b**) liver, (**c**) kidney, and (**d**) spleen, all of which were stained using hematoxylin and eosin (H&E). Inverted microscopy was used to determine the pathological anomalies at a magnification of ×400, with a scale bar of 50 µm. The red arrow indicates apoptotic cells, the black arrow denotes areas of necrosis, the blue arrow signifies binucleation, the green arrow illustrates hypertrophy, and the star represents cell death. The black boxes indicate areas of the enlarged views below. Scale bar = 50 µm.

**Table 1 pharmaceuticals-18-00960-t001:** In silico histone deacetylase inhibitory activity of CU4c.

Compound	HDAC Class I								HDAC Class II					
	HDAC1		HDAC2		HDAC3		HDAC8		HDAC4		HDAC6		HDAC7	
	∆*G*	K*_i_*	∆*G*	K*_i_*	∆*G*	K*_i_*	∆*G*	K*_i_*	∆*G*	K*_i_*	∆*G*	K*_i_*	∆*G*	K*_i_*
CU4c	−6.30	23.90	−6.17	29.85	−6.79	10.49	−5.89	47.81	−7.90	1.63	−7.00	7.43	−7.92	1.56

**Table 2 pharmaceuticals-18-00960-t002:** Selectivity Index (SI) of CU4c.

Compound	Exposure Times (h)	IC_50_ of CU4c, µM (Mean ± SEM)		SI
		A549 Cells	Vero Cells	
CU4c	24	93.90 ± 1.84	>200	2.13
	48	60.43 ± 2.18	>200	3.31
	72	40.34 ± 1.20	>200	4.96

The data represent the results of three different trials. SI = IC_50_ against Vero cells/IC_50_ against A549 cells. An SI value over 2 indicates preferential toxicity towards cancer cells [27].

**Table 3 pharmaceuticals-18-00960-t003:** CI and DRI of the combination regimens of CU4c, CDDP, and Gem against human lung cancer A549 cells.

Exposure Time	Parameters Used for Drug Interaction Assessment				CI	DRI		
	IC_50_ of CU4c (µM)		Subtoxic Dose of CDDP (µM)	Subtoxic Dose of Gem (µM)		CDDP	Gem	CU4c
	Alone	Combination						
			IC_20_	IC_20_				
24 h	93.90 ± 1.84	5.66 ± 0.25	21.00 ± 0.48	-	0.38 ± 0.00	3.13	-	16.59
48 h	60.43 ± 2.18	34.36 ± 0.81	7.12 ± 1.07	-	1.03 ± 0.01	1.91	-	1.76
72 h	40.34 ± 1.20	30.15 ± 3.13	1.69 ± 0.19	-	1.23 ± 0.10	3.22	-	1.34
48 h	60.43 ± 2.18	18.92 ± 1.97	-	0.68 ± 0.13	0.26 ± 0.02	-	16.00	3.19
72 h	40.34 ± 1.20	16.82 ± 0.42	-	0.35 ± 0.14	0.31 ± 0.02	-	3.83	2.40
			IC_30_	IC_30_				
24 h	93.90 ± 1.84	2.78 ± 0.11	32.73 ± 0.30	-	0.53 ± 0.00	2.01	-	33.78
48 h	60.43 ± 2.18	41.86 ± 2.23	9.16 ± 1.71	-	1.35 ± 0.04	1.48	-	1.44
72 h	40.34 ± 1.20	40.53 ± 3.03	2.68 ± 0.17	-	1.90 ± 0.11	2.03	-	1.00
48 h	60.43 ± 2.18	16.82 ± 1.52	-	1.30 ± 0.18	0.76 ± 0.01	-	8.37	3.59
72 h	40.34 ± 1.20	15.35 ± 3.63	-	0.55 ± 0.14	0.91 ± 0.12	-	2.44	2.63

**Table 4 pharmaceuticals-18-00960-t004:** Body weight, % body weight change (%BWC), and relative organ weight of mice in the vehicle control and treated groups.

Groups	Initial Body Weight (g)	Final Body Weight (g)	%BWC	Organ Index (g/100 g Body Weight)		
				Liver	Spleen	Kidney
Vehicle control	25.70 ± 0.76	28.18 ± 0.89	9.66	7.00 ± 0.39	0.64 ± 0.09	0.98 ± 0.01
CU 30 mg/kg	24.91 ± 0.17	27.73 ± 0.63	11.36	7.22 ± 0.21	0.62 ± 0.24	1.01 ± 0.01
CU4c 30 mg/kg	24.09 ± 0.88	27.52 ± 1.42	14.17	7.09 ± 0.44	0.70 ± 0.04	0.96 ± 0.07
Gem 50 mg/kg	24.86 ± 0.55	25.69 ± 0.54	3.36 *	7.31 ± 0.42	1.60 ± 0.08 *	1.00 ± 0.01
Gem 50 mg/kg + CU4c 30 mg/kg	23.63 ± 0.07	26.23 ± 0.55	11.01	7.32 ± 0.71	1.59 ± 0.35 *	1.11 ± 0.03

Data are expressed as means ± SD for each mice group: vehicle control, n = 5; 30 mg/kg CU, n = 4; 30 mg/kg CU4c, n = 5; 50 mg/kg Gem, n = 4; 50 mg/kg Gem + 30 mg/kg CU4c, n = 5. * *p* < 0.05 indicates a statistically significant change compared to the vehicle control treatment.

## Data Availability

The datasets produced and/or analyzed during this study may be obtained from the corresponding author upon a reasonable request.

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
