# Peer review of "Curcumin Derivative CU4c Exhibits HDAC-Inhibitory and Anticancer Activities Against Human Lung Cancer Cells In Vitro and in Mouse Xenograft Models"

_pharmaceuticals, 2025, doi:10.3390/ph18070960_

Round 1
Reviewer 1 Report
Comments and Suggestions for Authors
After a minor revision, the well-written article "Curcumin Derivative CU4c Exhibits HDAC Inhibitory and Anticancer Activities against Human Lung Cancer Cells in vitro and in Mouse Xenograft Models" should be published because it contains extensive research on the suggested topic.
-both the Abstract and the Conclusions could be improved by reporting data, IC50, IC30, IC20 values, percentages, binding energies, drug interaction, toxicological data, etc, from the research conducted, which highlights the efficiency of the research
- the Q3 values (Figure 9a) ​​are quite difficult to distinguish
- the authors do not mention whether such research on similar or more complex Curcumin derivatives is reported in the literature, and how good the results of this research are compared to those previously reported
Author Response
Reviewer 1
After a minor revision, the well-written article "Curcumin Derivative CU4c Exhibits HDAC Inhibitory and Anticancer Activities against Human Lung Cancer Cells in vitro and in Mouse Xenograft Models" should be published because it contains extensive research on the suggested topic.
-both the Abstract and the Conclusions could be improved by reporting data, IC50, IC30, IC20values, percentages, binding energies, drug interaction, toxicological data, etc, from the research conducted, which highlights the efficiency of the research
Response: The IC values are not mentioned in the abstract due to the 250-words limit; however, some IC50 values are mentioned in section 5 (Conclusions) (red letters) of the revised manuscript.
- the Q3 values (Figure 9a) ​​are quite difficult to distinguish
Response: We have modified Figures 9a and 10a in the revised manuscript to make the Q3 values more distinguishable.
- the authors do not mention whether such research on similar or more complex Curcumin derivatives is reported in the literature, and how good the results of this research are compared to those previously reported
Response: We have included the effects of other curcumin derivatives on lung cancer cells in sections 1 (lines 85-92) and 3 (lines 492-495) (red letters) of the revised manuscript.
Reviewer 2 Report
Comments and Suggestions for Authors
The paper, "Curcumin Derivative CU4c Exhibits HDAC Inhibitory and Anticancer Activities against Human Lung Cancer Cells in vitro and in Mouse Xenograft Models," is a well-written and comprehensive study that provides convincing evidence for the potential of CU4c in lung cancer therapy. The authors carefully investigated its HDAC inhibitory activity, antiproliferative effects, and potential for synergistic effects with gemcitabine and cisplatin in vitro and in vivo. However, despite the high overall quality of the paper, I have some fundamental concerns about certain aspects of methodology and reporting that must be addressed to ensure the scientific rigor and reproducibility of the manuscript before it can be considered for acceptance.
- This study focuses on the curcumin derivative CU4c. Explain why the same comprehensive experiments (e.g., cell cycle analysis, apoptosis induction, protein Western blot analysis, and especially in vivo xenograft models) were not performed on the parent compound, curcumin, in parallel with CU4c
- The paragraph on the molecular docking study in the Results section should be moved to its own subsection. It starts with "Moreover, a molecular docking study..." and ends with "...respectively (Figure 3g)." The current placement of "2.1. HDAC Inhibitory Activity of the Curcumin Derivative CU4c" ruptures up the flow of its narrative and the logical presentation of the experimental (in vitro) results. This sudden addition of computational data, along with its accompanying graphic (graphic 3g) and maybe other figures/tables that are connected, interrupts the planned flow of the subsection. Making a new subsection, like "2.X. Molecular Docking Analysis of CU4c with HDAC Isoforms," will make the Results section much clearer, better organized, and more scientifically sound.
- The purity or grade of all key chemicals and reagents (e.g., cell culture reagents, assay components, small molecules, etc.) used in the bioassay experiments must be clearly stated in the Materials and Methods section. Bioassay experiments are highly sensitive to small changes in reagent concentrations or the presence of impurities, which can significantly alter experimental results and affect reproducibility. Please specify the purity of each relevant compound as provided by the supplier (e.g., "analytical grade," "≥98% pure"). This detail is critical to ensure the reliability and validity of the reported results.
- The authors describe the extraction and isolation of curcumin from turmeric root. Curcumin naturally co-occurs with its closely related derivatives, demethoxycurcumin and bis-demethoxycurcumin, which are challenging to fully separate due to their similar structures and varying chemical stabilities. Anyway, the authors provide an HPLC image of CU and CU4c in the supplementary data. They should also refer to specific spectral data in the main text (e.g., Figure S5), allowing readers to easily consult the supplementary materials. Although the statement regarding spectral data are included in the Supplementary Materials section, it is important that the authors clearly reference them in the main text at the appropriate discussion points.
- The manuscript reports that curcumin is isolated from turmeric root. Although curcumin (and possibly its derivative, CU4c) of a specified purity is widely available commercially, provide specific reasons for extracting and isolating the curcumin in-house rather than purchasing it from a commercial source. This may include reasons such as cost-effectiveness at the required scale, specific purity considerations (e.g., presence of other curcuminoids), interest in the unique properties of the turmeric source, or optimization of the extraction process itself as part of the research
- Authors should mention the visualization method (e.g., UV light, chemical stain) and the characteristic used for identification (e.g., Rf values, color) for the TLC step in identifying DT2-1 and DT2-2, to improve reproducibility.
- The stated molar amount for curcumin (1.016 mg: 2.7580 mmol) appears to have a significant calculation discrepancy. Recheck this value, as it will fundamentally impact all subsequent stoichiometric calculations and reaction concentrations.
- Include the melting/decomposition point of the synthesized compound, (1E,4Z,6E)-1,7-bis(4-ethoxy-3-methoxyphenyl)-5-hydroxyhepta-1,4,6-trien-3-one (CU4c), in section 4.4, "Curcumin derivative CU4c Synthesis.
- Authors should mention the specific versions of AutoDock and GaussView used. In addition, ensure that proper citations are provided for AutoDock, GaussView, and Gaussian 03W, and that the sources of these software packages are cited.
- Include a comparison of this study's findings with Reference [20] in the introduction, particularly since it also reports a curcumin derivative. If Reference [20] is from your research group, explicitly state that it represents "our previous work" or use similar phrasing.
Author Response
Reviewer 2
The paper, "Curcumin Derivative CU4c Exhibits HDAC Inhibitory and Anticancer Activities against Human Lung Cancer Cells in vitro and in Mouse Xenograft Models," is a well-written and comprehensive study that provides convincing evidence for the potential of CU4c in lung cancer therapy. The authors carefully investigated its HDAC inhibitory activity, antiproliferative effects, and potential for synergistic effects with gemcitabine and cisplatin in vitro and in vivo. However, despite the high overall quality of the paper, I have some fundamental concerns about certain aspects of methodology and reporting that must be addressed to ensure the scientific rigor and reproducibility of the manuscript before it can be considered for acceptance.
1. This study focuses on the curcumin derivative CU4c. Explain why the same comprehensive experiments (e.g., cell cycle analysis, apoptosis induction, protein Western blot analysis, and especially in vivo xenograft models) were not performed on the parent compound, curcumin, in parallel with CU4c.
Response: In the previous study [20], we have already investigated the in vitro (cell cycle analysis, apoptosis induction, protein Western blot analysis) anticancer activity of curcumin prototype in parallel with the curcumin derivative CU17 in A549 lung cancer cells. In this study, we have investigated the anticancer activity of the parent compound, curcumin (CU), in vivo xenograft models in parallel with the curcumin derivative CU4c against A549 lung cancer cells (Figure 11).
2. The paragraph on the molecular docking study in the Results section should be moved to its own subsection. It starts with "Moreover, a molecular docking study..." and ends with "...respectively (Figure 3g)." The current placement of "2.1. HDAC Inhibitory Activity of the Curcumin Derivative CU4c" ruptures up the flow of its narrative and the logical presentation of the experimental (in vitro) results. This sudden addition of computational data, along with its accompanying graphic (graphic 3g) and maybe other figures/tables that are connected, interrupts the planned flow of the subsection. Making a new subsection, like "2.X. Molecular Docking Analysis of CU4c with HDAC Isoforms," will make the Results section much clearer, better organized, and more scientifically sound.
Response: Thank you for your constructive comments. We agreed and have moved the results of the molecular docking analysis of CU4c into a new subsection entitled 2.2. HDAC Inhibitory Activity of the Curcumin Derivative CU4c In Silico. The orders of the remaining subsections are changed accordingly.
3. The purity or grade of all key chemicals and reagents (e.g., cell culture reagents, assay components, small molecules, etc.) used in the bioassay experiments must be clearly stated in the Materials and Methods section. Bioassay experiments are highly sensitive to small changes in reagent concentrations or the presence of impurities, which can significantly alter experimental results and affect reproducibility. Please specify the purity of each relevant compound as provided by the supplier (e.g., "analytical grade," "≥98% pure"). This detail is critical to ensure the reliability and validity of the reported results.
Response: We have included the purity or grade of key chemicals in sections 4.1 (lines 628-643) (red letters) of the revised manuscript.
4. The authors describe the extraction and isolation of curcumin from turmeric root. Curcumin naturally co-occurs with its closely related derivatives, demethoxycurcumin and bis-demethoxycurcumin, which are challenging to fully separate due to their similar structures and varying chemical stabilities. Anyway, the authors provide an HPLC image of CU and CU4c in the supplementary data. They should also refer to specific spectral data in the main text (e.g., Figure S5), allowing readers to easily consult the supplementary materials. Although the statement regarding spectral data are included in the Supplementary Materials section, it is important that the authors clearly reference them in the main text at the appropriate discussion points.
Response: We have included the sentence..“The purities of CU4c and CU were provided in the supplementary materials (Figures S4 and S5, respectively”..in section 4.3 (lines 669-671) (red letters). Additionally, the sentence..“The NMR spectra and HPLC chromatograms of CU4c were provided in supplementary materials (Figures S1–S4)”..has been added in section 4.4 (lines 684-686) (red letters).
5. The manuscript reports that curcumin is isolated from turmeric root. Although curcumin (and possibly its derivative, CU4c) of a specified purity is widely available commercially, provide specific reasons for extracting and isolating the curcumin in-house rather than purchasing it from a commercial source. This may include reasons such as cost-effectiveness at the required scale, specific purity considerations (e.g., presence of other curcuminoids), interest in the unique properties of the turmeric source, or optimization of the extraction process itself as part of the research.
Response: The naturally derived histone deacetylase inhibitors possessing anticancer activity have been our research interest. Curcumin (CU) was previously extracted and isolated in a gram scale in our laboratory from abundantly available local turmeric using alternative solvents for the extraction (reported in a research article below). In addition, various curcumin derivatives were yielded by minor structural modification. However, the reason why we use CU isolated from turmeric root is mainly because of its availability in our lab. Therefore, we have no specific reason to mention in this study.
A previous research article: Kisanthia, R.; Hunt, A.; Sherwood, J.; Somsakeesit, L.-O.; Phaosiri, C. Impact of conventional and sustainable solvents on the yield, selectivity, and recovery of curcuminoids from turmeric. ACS Sustainable Chemistry & Engineering, 2022, 10 (1), 104-111.
6. Authors should mention the visualization method (e.g., UV light, chemical stain) and the characteristic used for identification (e.g., Rf values, color) for the TLC step in identifying DT2-1 and DT2-2, to improve reproducibility.
Response: We have included the sentence..“The reaction was monitored using TLC under UV light (λ=254 nm), then stained with an anisaldehyde solution for identification via Rf values and colorimetric analysis”..in section 4.3 (lines 663-665) (red letters).
7. The stated molar amount for curcumin (1.016 mg: 2.7580 mmol) appears to have a significant calculation discrepancy. Recheck this value, as it will fundamentally impact all subsequent stoichiometric calculations and reaction concentrations.
Response: We have rechecked and edited the molar amount for curcumin (1.016 g: 2.7580 mmol) with red letters in section 4.4 (line 674).
8. Include the melting/decomposition point of the synthesized compound, (1E,4Z,6E)-1,7-bis(4-ethoxy-3-methoxyphenyl)-5-hydroxyhepta-1,4,6-trien-3-one (CU4c), in section 4.4, "Curcumin derivative CU4c Synthesis.
Response: We have included the melting point of the synthesized CU4c in section 4.4 (line 684) (red letters).
9. Authors should mention the specific versions of AutoDock and GaussView used. In addition, ensure that proper citations are provided for AutoDock, GaussView, and Gaussian 03W, and that the sources of these software packages are cited.
Response: We have added the specific versions of AutoDock4 (V4.2.6) (line 700) and GaussView 3.0 (line 706) used for the molecular docking study in section 4.6 (red letters).
10. Include a comparison of this study's findings with Reference [20] in the introduction, particularly since it also reports a curcumin derivative. If Reference [20] is from your research group, explicitly state that it represents "our previous work" or use similar phrasing.
Response: We have added the sentence...“Furthermore, our previous research indicated that the curcumin derivative CU17 exhibited cytotoxic effects against NSCLC cells, demonstrating anti-cancer efficacy through G2/M phase arrest and apoptotic mechanisms [20].”...in the introduction section (line 90-92) (red letters).
Reviewer 3 Report
Comments and Suggestions for Authors
- In Figure 1, showing the synthesis of CU4c, how can the author confirm the exact structure of the product? Please explain using mass spectrometry and 1D-NMR data., and please analyze NMR data và add it to the result.
- What are the optimal concentrations of CU4c when combined with Gem or CDDP to achieve maximal synergistic effects without causing significant toxicity?
- What is the side effect of CU4c?
- What is the difference in treatment efficacy between the A549 cell line and other xenograft models? Please compare it with published references.
- What are the potential drug-drug interactions between CU4c and standard chemotherapy agents?
- How about anti-migration and anti-invasion? They are crucial in cancer therapy because they prevent tumor cells from spreading to the organs
Author Response
Reviewer 3
1. In Figure 1, showing the synthesis of CU4c, how can the author confirm the exact structure of the product? Please explain using mass spectrometry and 1D-NMR data., and please analyze NMR data và add it to the result.
Response: The compound CU4c was synthesized by reacting curcumin with K2CO3 and ethylbromide under reflux conditions as previously described (Ref. 26). Furthermore, the purified compound CU4c was identified through spectroscopy techniques, and its melting point was also determined. Its spectroscopic data was consistent with the previous reports (Ref. 26 & Ref. B below). Therefore, no mass spectrometry data was further obtained. The NMR data are provided as supplementary figures (Figure S1 and Figure S2). Analyzing chemical structure of CU4c using NMR data has been reported previously (Ref. 26).
Ref. B: Changtam, C.; Koning H. P.; Ibrahim, H.; Sajid, M. S.; Gould, M. K.; Suksamrarn, A. “Curcuminoid analogs with potent activity against Trypanosoma and Leishmania species” Eur. J. Med. Chem. 2010, 45, 941
2. What are the optimal concentrations of CU4c when combined with Gem or CDDP to achieve maximal synergistic effects without causing significant toxicity?
Response: The optimal doses of CU4c combined with CDDP at IC20 and IC30 were 5.66 and 2.78 μM, respectively. In contrast, the optimal doses when combined with Gem at IC20 and IC30 were 18.92 and 16.82 μM, respectively, as demonstrated in Table 3 (page 8).
3. What is the side effect of CU4c?
Response: According to the results in this study, CU4c exhibited a minimal impact on the growth of noncancerous Vero cells (IC50 >200 µM at 24, 48, and 72 h), as shown in Figure 4 (page 7). Additionally, it did not affect the visceral organs of nude mouse xenograft models (in vivo), as illustrated in Figure 12 (page 19).
4. What is the difference in treatment efficacy between the A549 cell line and other xenograft models? Please compare it with published references.
Response: Based on our knowledge, this research represents the first investigation of the curcumin derivative CU4c in lung cancer xenograft models A549. However, other cancer types, such as cervical and breast cancers, have not yet been studied with this compound. Therefore, we currently lack published references.
5. What are the potential drug-drug interactions between CU4c and standard chemotherapy agents?
Response: This study demonstrated that CU4c enhances the anticancer effects of Gem and CDDP. Consequently, the combination of these drugs may lead to greater effectiveness and reduced toxicity.
Synergistic Effects: The pairing of CU4c with either Gem or CDDP may produce a stronger effect than when either drug is used alone. This combination has the potential to improve treatment outcomes.
Reduced Individual Doses: By combining drugs, it employs reduced dosages of each drug while still achieving the intended therapeutic effect. This may help mitigate negative effects of each drug.
6. How about anti-migration and anti-invasion? They are crucial in cancer therapy because they prevent tumor cells from spreading to the organs.
Response: We agree with the reviewer that anti-migration and anti-invasion strategies are crucial for preventing tumor cells from spreading to the organs. However, in this study, we could not perform the tests due to our limited research funding. However, we have conducted more relevant, dependable, and suitable tests to investigate the anticancer effects of the drugs in nude mice xenograft models (in vivo). Our findings indicated that the CU4c combination therapy caused a significant decrease in tumor size, indicating that the CU4c treatment may impede cancer cell growth and dissemination, as demonstrated in Figure 11 (page 17).